

# Archival processes of the water stable isotope signal in East Antarctic ice cores

Mathieu Casado[1,2], Amaelle Landais[1], Ghislain Picard[3], Thomas Münch[4,5], Thomas Laepple[4], Barbara Stenni[6], Giuliano Dreossi[6], Alexey Ekaykin[7,8], Laurent Arnaud[3], Christophe Genthon[3], Alexandra Touzeau[1], Valerie Masson-Delmotte[1], and Jean Jouzel[1]

[1]Laboratoire des Sciences du Climat et de l'Environnement - IPSL, UMR 8212, CEA-CNRS-UVSQ-UPS, Gif sur Yvette, France
[2]Université Grenoble Alpes / CNRS, LIPHY, F-38000 Grenoble, France
[3]Université Grenoble Alpes / CNRS, LGGE, 38400 Grenoble, France
[4]Alfred Wegener Institute Helmholtz Centre for Polar and Marine Research, Telegrafenberg A43, 14473 Potsdam, Germany
[5]Institute of Physics and Astronomy, University of Potsdam, Karl-Liebknecht-Str. 24/25, 14476 Potsdam, Germany
[6]DAIS, Ca'Foscari University of Venice, Venice, Italy
[7]Arctic and Antarctic Research Institute, St. Petersburg, Russia
[8]Institute of Earth Sciences, St. Petersburg State University, Russia

*Correspondence to:* Mathieu Casado (mathieu.casado@gmail.com)

**Abstract.** The oldest ice core records are obtained from the East Antarctic plateau. Water isotopes records are key to reconstructing past climatic conditions over the ice sheet and at the evaporation source. The accuracy of climate reconstructions depends on knowledge of all the processes affecting water vapour, precipitation and snow isotopic compositions. Fractionation processes are well understood and can be integrated in Rayleigh distillation and isotope enabled climate models. However, a quantitative understanding of processes potentially altering the snow isotopic composition after the deposition is still missing. In low accumulation sites, such as those found in Antarctica, these poorly constrained processes are likely to play a significant role and limit the interpretation of isotopic composition.

Here, we combine observations of isotopic composition in the vapour, the precipitation, the surface snow and the buried snow from Dome C, a deep ice core site on the East Antarctic Plateau. At the seasonal scale, we suggest a significant impact of metamorphism on surface snow isotopic signal compared to the initial precipitation signal. Particularly, in summer, exchanges of water molecules between vapour and snow are driven by the sublimation/condensation cycles at the diurnal scale. Using highly resolved isotopic composition profiles from pits in five Antarctic sites, we identify common patterns, despite different accumulation rates, which cannot be attributed to the seasonal variability of precipitation. Altogether, the difference in the signals observed in the precipitation, surface snow and buried snow isotopic composition constitute evidences of post-deposition processes affecting ice core records in low accumulation areas.





## 1 Introduction

Ice is a natural archive of past climate variations. The chemical and physical compositions of ice, and the air bubbles trapped inside it, are used as paleoclimate proxies (*Jouzel and Masson-Delmotte*, 2010). Over large ice sheets, the water isotopes in ice cores can provide reconstructions of historic temperatures as far back as the last glacial period in West Antarctica (up to 60

000 years ago), (*WAIS Divide Project members*, 2013), and the last interglacial period in Greenland (120,000 years ago) (*North Greenland Ice Core Project members*, 2004; *NEEM Community members*, 2013). In East Antarctica, low accumulation rates enable the reconstruction of past climates over several interglacial periods, e.g. 420 000 years at Vostok (*Petit et al.*, 1999), 720 000 years at Dome F (*Kawamura et al.*, 2017) and 800 000 years at Dome C (*EPICA*, 2004, 2006). Even though reconstructions from ice cores from Greenland and West Antarctica do not extend as far back as from East Antarctica, high resolution

analyses of these cores provide very fine temporal resolution from which the seasonal cycle can be resolved (*Vinther et al.*, 2010; *Markle et al.*, 2017). Seasonal variations are also imprinted in the snow isotopic composition of high accumulation sites in coastal areas of Antarctica (*Morgan*, 1985; *Masson-Delmotte et al.*, 2003; *Küttel et al.*, 2012). For low accumulation sites as found on the East Antarctic Plateau, there is no consensus whether ice core records can reveal the climatic signal at resolutions finer than multidecadal (*Baroni et al.*, 2011) or not (*Ekaykin et al.*, 2002; *Pol et al.*, 2014; *Münch et al.*, 2016). *Ekaykin et al.*

(2002) analysed multiple pits from Vostok and identified large spatio-temporal variations caused by post-deposition processes associated with surface topography and wind interactions. These phenomena create strong non-climate variability in a single core, which calls for stacking isotopic composition profiles from several snow pits to reveal the common climatic signal. Still, on the East Antarctic Plateau, a clear seasonal cycle is depicted in the isotopic composition of the precipitation (*Fujita and Abe*, 2006; *Landais et al.*, 2012; *Stenni et al.*, 2016) and of the surface snow (*Touzeau et al.*, 2016). So far, whether this seasonal

cycle is archived or not in buried snow, and thus, whether stacking an array of snow pits enables us to increase the signal to noise ratio and depict a climatic record at the seasonal scale from water isotopic signal is an important open question (*Ekaykin et al.*, 2014; *Altnau et al.*, 2015; *Münch et al.*, 2016, 2017).

Several studies have focused on understanding how the climatic signal is archived in the isotopic composition of snow and

ice on the East Antarctic Plateau. Since the early works of *Dansgaard* (1964) and *Lorius et al.* (1969), the relationship between ice isotopic composition and local temperature has been attributed to the distillation associated with the successive condensation events on the path from the initial evaporation site to the deposition site (*Criss*, 1999). Nevertheless, the relationship between isotopic composition and surface temperature is not constant through time and space, due notably to processes within the boundary layer (*Krinner et al.*, 1997), the seasonality of the precipitation between glacial and interglacial periods (*Sime*

*et al.*, 2009), variations in air mass transport trajectories (*Delaygue et al.*, 2000; *Schlosser et al.*, 2004) and in the evaporation conditions (*Vimeux et al.*, 1999). For East Antarctica, the glacial-interglacial isotope-temperature relationship appears quite close to the spatial gradient, but its validity for inter-annual variations (*Schmidt et al.*, 2007) and warmer than present-day conditions (*Sime et al.*, 2009) is unclear.





In addition, under the exceptionally cold and dry conditions of East Antarctic drilling sites, the contribution of post-deposition processes to the isotopic composition of the snow cannot be neglected at the atmosphere/snow interface (*Town et al.*, 2008; *Sokratov and Golubev*, 2009). Indeed, the relationship between temperature and isotopic composition of the surface snow is different from the one found in the precipitation (*Touzeau et al.*, 2016). It has been recently evidenced that

summer exchanges between snow and water vapour at the surface significantly affect the isotopic composition of the snow both in Greenland (*Steen-Larsen et al.*, 2014) and on the East Antarctic Plateau (*Ritter et al.*, 2016). In this study, we intend to evaluate the different contributions to the snow isotopic composition signal in order to explain how the climatic signal is archived in ice isotopic composition. Therefore, we used Dome C as an open air laboratory to study the different contributions to the surface snow isotopic composition, combining : (1) direct precipitation input, (2) blowing snow, (3) exchanges with the

atmospheric vapour and (4) exchange with the firn below the surface; as presented in Fig. 1.

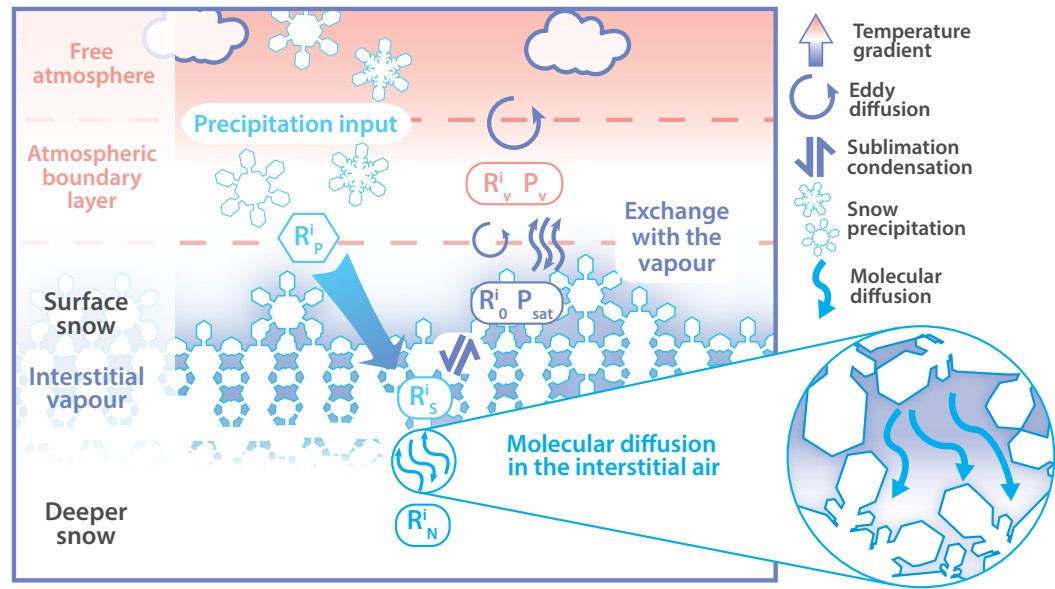

**Figure 1.** Schematic of the different contributions to the snow isotopic composition ($R^i_X$ stands for the composition of isotope $i$ in the phase $X$: $P$ stands for precipitation, $V$ stands for vapour, $S$ stands for surface snow, $N$ stands for the deeper snow, 0 stands for the vapour at equilibrium with the snow): above the surface, both the precipitation and the sublimation/condensation cycles can contribute to the surface composition; in the open-porous firn below the surface, ice crystals can exchange with the air in the pores, influenced or not by wind pumping. Deeper in the firn, molecular diffusion in the interstitial air affects the snow isotopic composition.

At a site as far along the distillation path as Dome C (and in general on the East Antarctic Plateau), only a small amount of precipitable water is available (*Ricaud et al.*, 2014) and precipitation is sparse and irregular (*Genthon et al.*, 2015). As a result, accumulation of snow at Dome C is not a homogeneous process affecting a flat surface, and the snow deposition is patchy

and strongly dependant of the surface roughness (*Groot Zwaaftink et al.*, 2013; *Libois et al.*, 2014; *Picard et al.*, 2016a). As





a result, a small but significant contribution to the annual mass balance comes from sublimation/condensation directly at the surface (*Genthon et al.*, 2017). Indeed, during the Austral summer, negative fluxes up to $-0.3\ mm\ w.e.\ month^{-1}$, associated with sublimation, are observed whereas during the Austral winter, positive fluxes up to $0.1\ mm\ w.e.\ month^{-1}$, associated with condensation, are found. Integrated, these contributions for a site such as Dome C add up to 10% of the total annual

accumulation. Even though the amount of water introduced by this contribution is limited, we expect a significant impact on the isotopic budget of the snow.

At the diurnal scale, this asymmetry of temperature conditions during condensation/sublimation cycles is also observed in summer: even though the sun never goes down, the radiative budget creates a significant temperature cycle (up to $16\,^{\circ}C$ per day)

during which the vapour isotopic composition evolves in parallel with the snow surface temperature (*Casado et al.*, 2016; *Ritter et al.*, 2016). As sublimation occurs at higher temperature than condensation, the diurnal cycle results in a non-neutral isotopic budget. In addition, the dynamic of the atmospheric boundary layer is different during the day and during the night: during the day, active convection enables turbulent mixing throughout the boundary layer while during the night, the conditions are much more stratified as illustrated by the important temperature gradients that are observed (*Casado et al.*, 2016; *Vignon et al.*, 2017).

In the top first metres of the snowpack, isotope exchanges involved during snow metamorphism and diffusion within the porous matrix additionally affect the isotopic composition of the snow (*Langway*, 1970; *Johnsen*, 1977; *Whillans and Grootes*, 1985; *Calonne et al.*, 2015). The diffusion length associated with these processes depends on the firn ventilation, the snow density, porosity and tortuosity and the exchange rate between the atmospheric water vapour and the surface snow (*Johnsen*

*et al.*, 2000; *Gkinis et al.*, 2014). This wide range of processes hampers the interpretation of the isotopic signal. In particular it is not clear how much of the original signal acquired during the formation of the precipitation is conserved during the burial of the snow (*Münch et al.*, 2017).

Recent studies have focused on the monitoring of the isotopic composition of the snow pack on the East Antarctic Plateau

(*Touzeau et al.*, 2016), of the precipitation (*Fujita and Abe*, 2006; *Landais et al.*, 2012; *Stenni et al.*, 2016), and of the atmospheric water vapour (*Casado et al.*, 2016; *Ritter et al.*, 2016); exploring their links to climatic parameters and the implications for post-deposition processes during the archiving of the climatic signal by the snow isotopic composition. Here, we study the isotopic composition of the continuum between atmospheric vapour, precipitation, surface and buried snow. To do so, we combine different datasets from Dome C, on the East Antarctic Plateau, based on published studies and new results, in order to

qualitatively characterise the different processes affecting surface snow isotope composition from diurnal to annual time scales. We first report and compare the different methodologies applied for sampling surface snow, snow pits, precipitation and water vapour in the atmosphere (Section 2). Then, we deliver the results from different studies including surface snow measurements over several years, precipitation measurements, vapour and snow measurements and snow pits sampling (Section 3) before summarising our key conclusions.




## 2  Sites, material and methods

### 2.1  Site

The East Antarctic Plateau is a high elevation area, over 2500 metres above sea level ($m\,a.s.l.$) covered with snow and ice
spreading across most of the eastern continental part of Antarctica (Fig. 2). The East Antarctic Plateau is characterised by
5   mean annual temperatures below $-30^\circ C$ and accumulation below $80\,kg.m^{-2}.yr^{-1}$, as illustrated in Fig. 2.

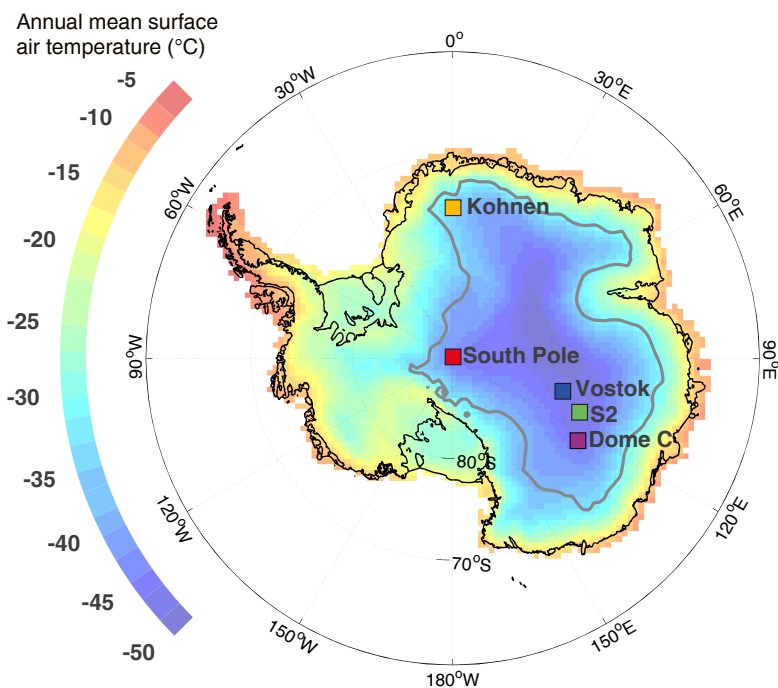

**Figure 2.** Map of Antarctica highlighting the East Antarctic Plateau (grey line = contour of $2500\,m$ a.s.l. elevation) indicating the location
of the sampling sites (solid squares) included in this work. Colours indicate the annual mean surface air temperature at $2\,m$ modified from
the ERA-interim dataset from 1979 to 2009 (*Nicolas and Bromwich*, 2014).

    This study mainly focuses on Dome C, the site of the permanent station Concordia, enabling year-long operations and
thus giving the rare opportunity to study the entire seasonal cycle of isotopic composition. This site is characterised by low
accumulation ($27\,mm\,w.e.$ per year) and low temperature ($-52.4^\circ C$). We present the conditions found at Dome C compared
10   to other deep ice core sites on the East Antarctic Plateau: Kohnen, Vostok and Amundsen-Scott South Pole stations, and the
point S2, which is one of the drilling sites of the campaign Explore-Vanish joining Dome C and Vostok (see Fig. 2). These
locations span a large range of climatic conditions of the East Antarctic Plateau as illustrated on Table 1.





**Table 1.** Climatic conditions at the different sites used in this study (*Alley*, 1980; *Petit et al.*, 1982; *Wendler and Kodama*, 1984; *Oerter et al.*, 2000; *Ekaykin et al.*, 2002; *van As et al.*, 2007; *Lazzara et al.*, 2012; *Casey et al.*, 2014; *Genthon et al.*, 2015; *Touzeau et al.*, 2016; *Laepple et al.*, 2016)

| Site | Location | Altitude (m a.s.l.) | AWS mean temperature (°C) | $10\,m$ firn temperature (°C) | Accumulation ($kg.m^{-2}.yr^{-1}$) | Mean wind speed ($m.s^{-1}$) |
|------|----------|---------|---------|---------|---------|---------|
| Kohnen | 75.0° S - 0.1° E | 2892 | -42.2 | -44.5 | 62 - 73 | 4.5 |
| Vostok | 78.5° S - 106.8° E | 3488 | -55.2 | -57 | 21 | 5.1 |
| S2 | 76.3° S - 120° E | 3229 | NA | -55.1 | 21 | NA |
| Dome C | 75.1° S - 123.3° E | 3233 | -52.4 | -54.3 | 27 | 3.3 |
| South Pole | 90° S - 0° E | 2835 | -49.3 | -50.8 | 80 | 4.1 |

## 2.2 Surface snow and precipitation sampling

Precipitation and surface snow have been regularly sampled at Dome C by different teams since 2008. Here, we report new measurements of precipitation and surface snow together with previously published data from *Stenni et al.* (2016) and *Touzeau et al.* (2016) (see Table 2). Because different teams were in charge of the different sampling activities, the protocols differ
5    between the years.

The sampling protocol of the 2011 campaign (SUNITEDC) has been precisely described by *Touzeau et al.* (2016): the upper first millimetres of snow (1 to $5\,mm$) were gathered every 1-2 weeks using a metallic blade over a surface of 20 by $20\,cm$. This leads to samples of approximately $20\,mL$. The sampling areas were randomly picked, provided the surface was flat.

During the NIVO project (from 2013 to 2016), the surface snow was gathered by sampling roughly $15\,mm$ of snow with a corning flask over a surface of 20 by $10\,cm$. This led to samples of approximately $50\,mL$. The sampling areas of the NIVO project were chosen randomly in a 100 by $100\,m$ "clean area" near the Atmospheric Shelter in parallel with density and specific surface area (SSA) measurements (see section 2.4). To address the problem of spatial variability, two samples separated by 10
15    to $50\,m$ were gathered during each collection and we present here the average value of the two samples. In addition, during summer 2013/14, regular samplings of surface and sub-surface snow were performed daily for almost 2 months. The surface samples were gathered using a corning flask from 0 to $3\,cm$ depth. The sub-surface samples were gathered by the same tool from 3 to $6\,cm$ depth. In 2014/15, an additional sampling took place within the GLACIOLOGIE project twice a day from December 2014 to January 2015 near the location of the inlet used for water vapour monitoring (See section 2.4 and *Casado*
20    *et al.* (2016)) following the same protocol ($15\,mm$ thick samples gathered directly in the snow).

Sampling of surface snow and isotopic composition of precipitation were carried out in parallel by the Italian winterover crews (program PRE-REC). Precipitation samples have been collected all year round on an 80 by $120\,cm$ wooden table stand-



ing $1\,m$ above the ground level $800\,m$ from Concordia Station from 2008 to 2011. The samples were collected at 1 a.m. UTC every day if the amount was sufficient. As already mentioned by *Stenni et al.* (2016), a few episodes of blowing snow might affect the precipitation gathered on the plate. The surface snow samples were gathered from an adjacent wooden plate of $80$ by $120\,cm$ at ground level. If the amount of snow on this second table was sufficient, snow samples were collected. Both

precipitation and surface snow samples were sealed into date-labelled plastic bags and preserved in a frozen state until delivery and measurement in Italy. For the precipitation samples, the protocol is detailed by *Stenni et al.* (2016). It is important to note that the protocol of surface snow sampling from the PRE-REC campaign differs greatly from the protocols from the NIVO and SUNITEDC programs due to the presence of the wood plate.

**Table 2.** Summary of the different campaigns of surface snow and precipitation samplings presented here.

| Project | Location | Years | Resolution (days) | Reference |
|---|---|---|---|---|
| SUNITEDC (French) | Surface snow | 2011 | 7 | (*Touzeau et al.*, 2016) |
| PRE-REC (Italian) | Precipitation | 2008 to 2011 | 1 | Partially in (*Stenni et al.*, 2016) |
| | Surface snow | 2012 and 2014 | 7 | This study |
| NIVO (French) | Surface snow | 2013 to 2016 | 3 | This study |
| | Sub-surface | 11/2013 to 01/2014 | 1 | This study |
| GLACIOLOGIE (French) | Surface snow | 12/2014 to 01/2015 | 1 | This study |

## 2.3 Snow pits sampling

We present profiles of isotopic composition sampled in snow pits at Dome C : two unpublished profiles from the first preliminary campaign at Dome C in 1978 and two new snow pit profiles obtained in 2014/15, dug $50\,m$ apart in parallel with surface snow sampling and vapour monitoring. For one of them, snow temperature and density profiles were established. The samples were taken in plastic flasks and analysed later on in the laboratory. To extend the results to other sites of the East Antarctic

Plateau, we compare the isotopic profiles to other snow pit samplings performed through several campaigns over different sites of East Antarctica which were realised and analysed by different teams.

In addition to the Dome C snow pits, two new isotopic composition profiles from Kohnen were extracted from trenches, following the methodology reported in *Münch et al.* (2016) but up to a depth of $3.6\,m$ and a vertical resolution of $3\,cm$. Both

profiles are separated by approximately $500\,m$ well beyond the decorrelation scale of the stratigraphic noise, to enable evaluation of the spatial variability in the profiles of isotopic composition. A large number of snow pits from Vostok station are presented here as well, which were previously described in *Ekaykin et al.* (2002, 2004) and *Ekaykin and Lipenkov* (2009). We combine the results from six snow pits with depths varying from $2.5\,m$ to $12\,m$ and a minimum resolution of $5\,cm$. In addition, snow pits from the Explore-Vanish campaign are included comprising one $3.5\,m$ deep snow pit from Vostok, one $2.6\,m$



**Table 3.** Summary of the different snow pits presented in this study.

| Station | Years | Resolution (cm) | Number of pits | Reference |
|---|---|---|---|---|
| Vostok | 2001 to 2015 | 2 to 5 | 6 | Ekaykin et al. (2002, 2004, 2009) |
| | 2012/13 | 3 | 1 | *Touzeau et al.* (2016) |
| Kohnen | 2014/15 | 3 | 2 | This study |
| Dome C | 1977/78 | 1 to 3 | 2 | This study |
| | 2012/13 | 3 | 1 | *Touzeau et al.* (2016) |
| | 2014/15 | 1.5 to 5 | 2 | This study |
| S2 | 2012/13 | 3 | 1 | *Touzeau et al.* (2016) |
| South Pole | 1978 | 2 | 1 | *Jouzel et al.* (1983) |
| | 1989/90 | 1.1 | 1 | *Whitlow et al.* (1992) |

deep from S2 and one $2\,m$ deep from Dome C, all of them including triple isotopic compositions ($\delta^{18}O$, $\delta^{17}O$ and $\delta D$) published in *Touzeau et al.* (2016). Finally, we include two snow pits from the South Pole (*Jouzel et al.*, 1983; *Whitlow et al.*, 1992).

We do not expect any impact of the sampling technique when comparing the different snow pits considering the similarity of the protocols of all the snow pit samplings.

## 2.4 Atmospheric and snow surface monitoring

Water vapour isotopic composition has been measured at Dome C in 2014/15 (*Casado et al.*, 2016). To reduce the noise, the dataset was averaged to hourly resolution and covers approximately one month. In parallel to water vapour isotopic composition monitoring, surface snow was sampled once to twice a day. For a period of 27 hours, the surface snow was sampled every hour to evaluate the diurnal cycle of both the vapour and the snow isotopic composition (see section 2.2).

We use temperature, wind speed and humidity measurements from the 45m meteorological profiling system described by *Genthon et al.* (2013). The temperature and humidity observations are performed using ventilated thermohygrometers HMP155 and are therefore free of radiation biases (*Genthon et al.*, 2011). The temperature reanalysis product (ERA interim) has been compared to ventilated automatic weather station data (AWS) from *Genthon et al.* (2013) and we found a good agreement at the seasonal scale and fairly good agreement at the event scale (not shown here). Wind speed and direction are measured using Young 05103 and 05106 aerovanes. Snow surface temperature is measured with a Campbell scientific IR120 infrared probe located $2\,m$ above ground level.

In the field, snow metamorphism is difficult to quantify due to the large noise created by the spatial variability, requiring a large number of samples every day. Therefore, we include grain index observations (*Picard et al.*, 2012) obtained by satellite measurements using passive microwave satellite data. *Picard et al.* (2012) argue that the grain index is an indicator of the





coarsening of snow grains and show its increase in summer to be anti-correlated with the integrated summer precipitation amount. When available, we include Surface Specific Area (SSA) measurements also as an indicator of metamorphism (*Libois et al.*, 2015). These optical methods are completed with snow observations. Frost deposition was monitored with a time lapse of the growth of hoar at the surface (see the video at https://vimeo.com/170463778). An image processing script was used to

characterise the growth of a few crystals at the surface of a sastruga.

## 2.5 Modelling approaches

To investigate the impact of post-deposition processes, it is necessary to present how the surface snow isotopic composition differs from the initial precipitation signal formed during the Rayleigh distillation. Here, we make use of the Rayleigh-type Mixed Cloud Isotope Model (MCIM) developed by *Ciais and Jouzel* (1994) which computes the Rayleigh distillation along

the air mass trajectories. The model includes microphysical properties of clouds and in particular takes into account mixed phase conditions. It is tuned with triple snow isotopic composition measured along a transect from Terra Nova Bay to Dome C (*Landais et al.*, 2008). One of the main uncertainties in this model is the supersaturation. Indeed, *Jouzel and Merlivat* (1984) evaluate the impact on kinetic fractionation during the snow formation parametrised from the supersaturation. The tuning of the supersaturation has been proven suitable to evaluate the variations of isotopic composition at Dome C (*Winkler*, 2012).

This will provide a comparison between the spatial (at the scales from 10 to $1000\,km$) and the temporal slope of the isotopic composition of precipitation at the seasonal scale. It will also be used to quantify the impact of post-deposition processes on the surface snow isotopic composition by providing a reference for the precipitation isotopic composition, for days with no precipitation.

## 3  Results and discussions

In this section, we review the results from the different datasets, illustrating the different steps of the archiving of the climatic signal by the snow isotopic composition, from the precipitation to the buried snow (see Fig. 1). As we expect the precipitation input to be the primary contribution, we will first compare the variations of precipitation isotopic composition based on data from *Stenni et al.* (2016) and additional new data, to surface snow isotopic composition. Then, we will evaluate the exchanges with the atmosphere: first at the diurnal scale by reporting parallel measurements of water vapour and surface snow isotopic

composition; and second, at the day-to-day and at the seasonal scales the difference between surface snow and precipitation isotopic composition to the meteorological conditions and the grain index. Finally, we will compare the isotopic composition of the surface, sub-surface and buried snow to evaluate the exchanges within the firn.





### 3.1 Precipitation and surface snow isotopic composition time series

#### 3.1.1 Precipitation isotopic composition

In this section, we present precipitation isotopic composition data at Dome C from *Stenni et al.* (2016) depicting 3 complete annual cycles from 2008 to 2010, completed by a new, unpublished dataset from 2011 (Fig. 3).

At Dome C, the isotopic composition of precipitation shows large variability at the day-to-day scale and a regular seasonal cycle. In summer, we systematically observe precipitation $\delta^{18}O_p$ above -40 ‰ , whereas in winter, $\delta^{18}O_p$ values below -65 ‰ are systematically observed. At the seasonal scale, the isotopic composition of precipitation is relatively well correlated to the local temperature with a slope of $0.46\,‰°C^{-1}$ ($R^2 = 0.65$, $n = 1111$), this slope is similar to the one obtained by *Stenni et al.*

(2016) for the years 2008 to 2010 of $0.49\pm0.02\,‰°C^{-1}$ ($R^2 = 0.63$, $n = 500$). No lag between temperature and isotopic composition variations is observed.

Compared to other year-long precipitation time series on the East Antarctic Plateau, this slope is lower than at Dome F (0.78 $‰°C^{-1}$ with $R^2 = 0.78$ (*Fujita and Abe*, 2006)) and higher than at Vostok ($0.26\,‰°C^{-1}$ with $R^2 = 0.58$ (*Touzeau et al.*,

15 2016)).

#### 3.1.2 Surface snow isotopic composition

Here, we present measurements of surface snow isotopic composition at Dome C from December 2010 to January 2016 (Fig. 3) combining results from *Touzeau et al.* (2016) with new data presented for the first time in this study from the PRE-REC, NIVO and GLACIO projects. The dataset includes three complete annual cycles of surface snow isotopic composition (in 2011, 2014

and 2015) and part of the 2012 cycle, with the respective temperature variations and the precipitation events (from reanalysis products, ERA-interim).

First, we focus on the impact of local spatial variability (below $1\,km$) of the measurements. Indeed, significant differences can be found in the snow isotopic composition for distances ranging from 1 to $1000\,m$. To disentangle the local spatial vari-

ability from the temporal variations of the surface snow isotopic composition (Fig. 3), we compare the duplicate measurements realised during the year 2014 in the project NIVO (red shade in Fig. 4 and see Section 2.2). We found that the spatial uncertainty of the surface snow isotopic composition measurements is 3.4‰ for $\delta^{18}O_s$ from 2 standard deviations calculated with the duplicates on the NIVO samples (randomly picked within $50\,m$). For that year, several sets of measurements are available from 3 different programs (PRE-REC in blue, NIVO in red and GLACIOLOGIE in black). Even though the PRE-REC sam-

pling is more affected by heavy snowfall events due to the wood plate, apart from a limited number of outliers (5 over 59), the dataset is most of the time representative of the surface snow and not of the precipitation. This provides an independent test to validate our estimation of the spatial variability. Strong differences are visible at the event scale, in particular during some events in March, May and June. During these events, the very low values of the PRE-REC sampling are in agreement





**Figure 3.** Monitoring of the precipitation and of the surface snow at Dome C from 2008 to 2016, from top to bottom: surface snow isotopic composition ($\delta^{18}O_s$, light green lines) from December 2010 to 2015: Dec 2010 to Dec 2011, data from the project SUNITEDC (*Touzeau et al.*, 2016); Feb 2012 to Oct 2012, data from the project PRE-REC (this study); Nov 2013 to Jan 2016, data from the project NIVO (This study), the light green shaded area is the uncertainty due to spatial variability obtained from replicates ($2\sigma$, see Section 3.1.2) and the dark green line is the modelled surface snow isotopic composition from the toy model detailed in section 3.2; four years (2008 to 2011) of monitoring at Dome C of the variations of the isotopic composition of precipitation from the PRE-REC campaign (*Stenni et al.*, 2016) ($\delta^{18}O_p$, blue, dots: raw data, line: monthly average), for comparison we present the grain indices from satellite observations (*Picard et al.*, 2012) (black fine line), and $2\,m$ temperature measurements (red line) and precipitation (black bars) from ERA-interim reanalysis product. The blue shaded areas highlight the high grain index values (arbitrary threshold on the variations indicating when metamorphism is active).

with the isotopic composition of precipitation (Dreossi, personal communication). Apart from these outliers, when comparing the PRE-REC results to the NIVO results, the average difference is 1.5‰, which we attribute to both spatial variability and mixture of precipitation and surface snow. The comparison of the NIVO and GLACIOLOGIE datasets shows much smaller differences (on average 0.4 ‰) but the comparison was done on a limited period without any large synoptic event. These in-





dependent sampling campaigns validate our uncertainty of the surface snow $\delta^{18}O_s$ of 3.4‰. Still, at the event scale (synoptic event of typically a couple of days), the variations of the surface snow isotopic composition exhibit an important small scale spatial variability (meter scale) due to the patchiness of the accumulation and snow redistribution. Caution in interpretation of variations of surface snow isotopic composition at short time scales is therefore necessary.

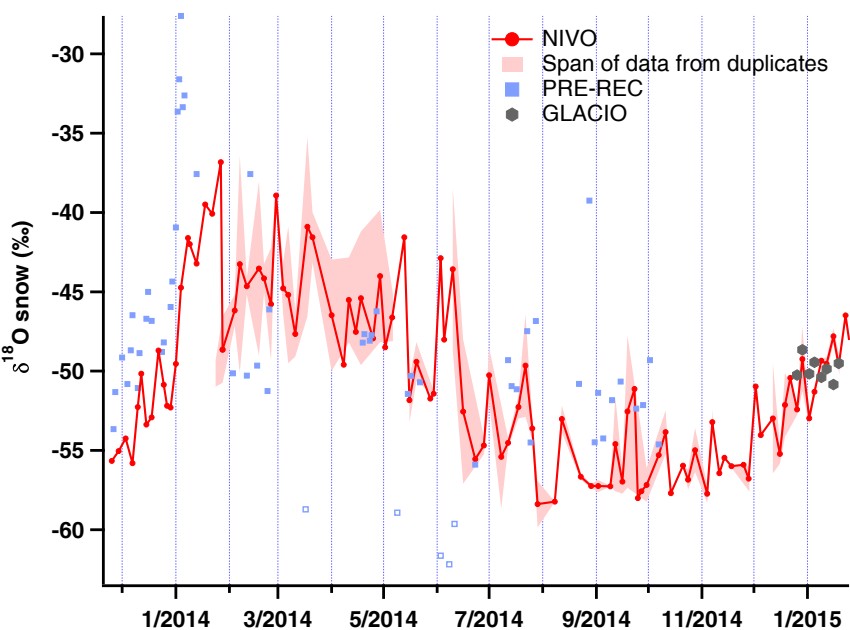

**Figure 4.** Reproducibility of surface snow isotopic composition measured in 2014. The NIVO dataset (red dots and line, duplicate span represented by the red shade, corresponding to the green line in Fig. 3) is compared to measurements obtained by independent teams (PRE-REC, blue squares; GLACIO, grey hexagon) and independent methods: PRE-REC snow samples were collected from a wood table whereas NIVO and GLACIO snow samples were collected directly from the surface snow. The outliers in the PRE-REC samples are highlighted by a white centre of the squares.

Second, we focus on the temporal variability at different time scales. The three years (2011, 2014 and 2015) present the typical temperature variations for the East Antarctic Plateau (see Fig. 3): a short, "warm" summer before a long, rather constant, winter as described by *Van Den Broeke* (1998). Over this cycle are imprinted short warm events often associated with advection of warmer air masses and precipitation; these warms events are particularly visible in winter. We observe a similar

10 pattern for $\delta^{18}O_s$ of the surface snow: annual cycles with a steep maximum centred on January (roughly a month after the temperature maximum) and a gradual decrease along most of the winter delayed by several months when compared to the local air temperature. Over these annual cycles, peaks of $\delta^{18}O_s$ of surface snow occur and some of them might be related to warm precipitation events as previously suggested by *Touzeau et al.* (2016).



The winter values of the surface snow isotopic composition are similar between all observed years but the summer values show strong inter-annual variations, resulting in variable amplitudes of the annual cycles of $\delta^{18}O_s$. In 2011 and 2015, the amplitude of the annual cycle is below 10‰ in $\delta^{18}O_s$, whereas it is above 20‰ in 2014. In 2012, despite missing data at the beginning of the year, we observe variations of $\delta^{18}O_s$ more similar to the ones of 2014 than to the ones of 2011 and 2015

with a difference of 15‰ in $\delta^{18}O_s$ between the maximum at the beginning of February and the minimum in September. These differences are significant with respect to the results obtained from replicate samples. Several hypotheses can be proposed to explain the variability in the summer increase of $\delta^{18}O_s$ of the surface snow: these include variability of the amount of snowfall during summer, which will be addressed in the next section, and post-deposition processes which will be evaluated in Section. 3.3.

### 3.2    Contribution of the precipitation to the surface snow isotopic composition

Considering the large variability in the amount of snow deposited during each precipitation event, we study how the cumulative contribution of each precipitation event associated with different $\delta^{18}O_p$ can influence the surface snow $\delta^{18}O_s$ and create a different signal from that observed in the precipitation. We use the slope isotope/temperature in the precipitation ($0.46\,‰°C^{-1}$,

*Stenni et al.* (2016)) to create a modelled snow fall isotopic composition product from ERA-interim temperature. The precipitation $\delta^{18}O_p$ generated with this method matches the amplitude of the seasonal variations observed from 2008 to 2011 (above -40 ‰ in summer and below -65 ‰ in winter) as well as the warm anomaly associated with the advection of moist air masses (see Fig. 11 in Supplementary Materials A). This modelled precipitation isotopic composition can be transferred in a surface snow isotopic composition by weighting the isotopic composition of each precipitation event with the amount of snow (see

Supplementary Materials A).

As the snow surface samples are $1.5\,cm$ thick, they contain the accumulation of several precipitation events (roughly the thickness in accumulation expressed in snow equivalent). To evaluate the impact of the mixing over a $1.5\,cm$ thickness, absolute values of the accumulate rate are important, and so we use precipitation estimates from ERA-interim products which have been renormalised in order to obtain a total amount of accumulation over 7 years matching the observations (*Genthon et al.*,

2015). The results of this modelled surface snow isotopic composition are presented in Fig. 3. We observe that the seasonal cycle of the modelled surface snow isotopic composition is in most cases delayed compared to the temperature seasonal cycle. In particular, the summer maximum of isotopic composition is observed roughly 2 months after the temperature high. We also observe that the modelled amplitudes of these summer excursions are not regular, some presenting high amplitudes (2011,

2012 and 2014 for instance) while others are capped at values below -50 ‰ (2015 and 2016 for instance). This is independent of the temperature seasonal cycle which is rather regular over this period. These two features can be explained by: (1) the surface snow signal, which integrates several weeks of precipitation, and should thus be delayed compared to the temperature seasonal cycle, and (2) the heavy snowfalls, which are associated with warm conditions, and lead to over representation of the





precipitation with high isotopic composition.

The model accurately reproduces some of the differences between the signal in the surface snow and in the precipitation: for instance, the temporal lag of the summer maximum of the surface snow compared to the precipitation. The model also explains why no $\delta^{18}O_s$ values below -65 ‰ are recorded in the winter surface snow. Indeed, the cold snow events are only associated with a small amount of precipitation and do not leave a significant imprint on the surface snow isotopic composition. However, the model does not capture the short-term variations of the surface snow isotopic composition, e.g. the very enriched surface snow isotopic compositions in June and August 2012 are not reproduced by the model. At the seasonal scale, we also observe some discrepancies: in winter, the calculated $\delta^{18}O_m$ matches the measured values, except during winter 2015 when the higher than usual surface snow isotopic composition are not captured by the model; in summer, the summer highs are accurately represented by the model in 2012 and 2014, whereas in 2011 and 2015, the model and data present important differences. The mismatch observed in summer 2015 can be linked to the rather low amount of precipitation: due to the lack of precipitation events (less than 0.3 mm w.e. cumulated from October 2014 to January 2015), the modelled $\delta^{18}O_m$ remains flat until the 17th of February 2015 whereas on-site samples indicate an increase of the surface snow isotopic from December 2014 on.

Our simulation did not consider the influence of the patchiness of the accumulation. We expect the post-deposition processes creating the patchiness of accumulation (for instance redistribution by the wind in interaction with the surface roughness) to be erased at the seasonal scale as it is a random process. Therefore it does not explain why values are systematically different over several weeks between the surface snow and the precipitation isotopic compositions. In the case of summer 2014, we obtained two time-series of surface snow isotopic composition, and monthly averages of snow isotopic composition do not show significant differences between the different datasets. In the case of summer 2015, a limited number of samples has been recovered independently from the NIVO project, which show a good agreement for the average value of the summer amplitude rise of $\delta^{18}O_s$. Additionally, for the NIVO and the SUNITEDC campaigns, the samples were randomly taken, and so should not be systematically affected by the erasing of the precipitation signal in the surface snow isotopic composition if it was linked to the patchiness of the accumulation.

## 3.3 Contribution of the exchanges between atmospheric water vapour and snow isotopic composition

### 3.3.1 Observations of a frost deposition event

We focus on the isotopic exchanges between the surface snow and the atmospheric vapour at the diurnal scale by comparing measurements of atmospheric vapour isotopic composition from *Casado et al.* (2016) with new results of surface snow isotopic composition from samples obtained in parallel with the vapour monitoring over 24 hours on the 7th of January 2015 (Fig. 5). The spatial variability of the surface snow isotopic composition was estimated by realising two types of duplicates: every hour, two samples were taken from a random location over a $30\,m^2$ field and one sample was taken at a fixed location. The error bar



presented in Fig. 5 represents the noise on the surface snow due to the spatial variability.

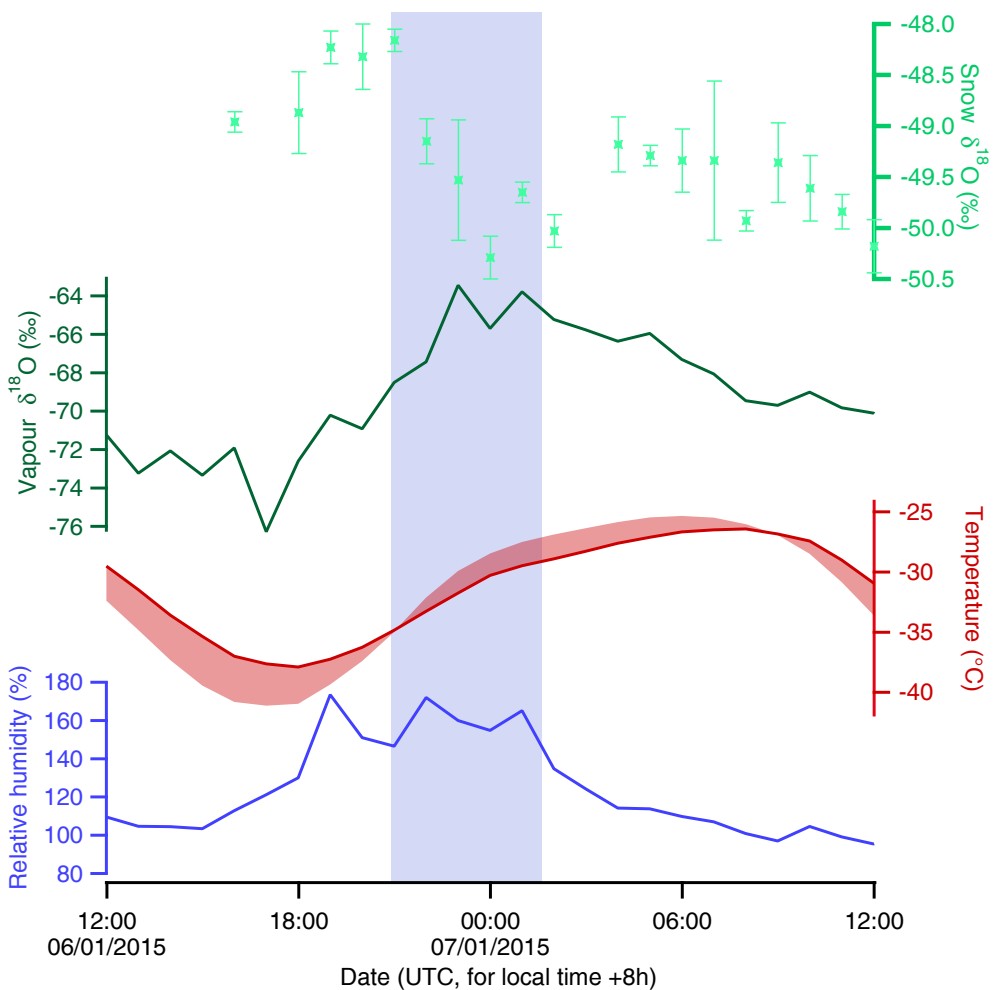

**Figure 5.** Isotopic composition of surface snow (light green dots, error bars are obtained from replicates) compared with isotopic composition of water vapour (dark green line) during one typical summer day at Dome C , with 3m-temperature (purple line) compared to the surface temperature (light purple shade) and to the relative humidity (blue line) during the same period from *Casado et al.* (2016). The blue shaded area marks the time-period when frost deposition was observed (see the time-lapse video: https://vimeo.com/170463778)

The 7th of January was characterised by a large diurnal cycle in water vapour isotopic composition, humidity and temperature associated with a turbulent and convective atmospheric boundary layer enabling important exchanges of moisture between snow and vapour (*Casado et al.*, 2016). This is a common situation in summer at Dome C due to weak katabatic winds. The afternoon was perturbed by a frontal system which affected the air vapour isotopic composition from 06:00 on the 7th of January (the maximum of $\delta^{18}O_v$ is typically reached around 06:00 UTC). To study the exchanges between snow and vapour



without being impacted by meteorological events, we therefore focus on the night from 18:00 on the 6th of January to 5:00 on the 7th of January. In addition to the isotopic composition of the vapour and of the surface snow, in Fig. 5 we present 3m temperature measured by AWS, surface temperature measured by infrared sensing (*Casado et al.*, 2016) and relative humidity calculated from the specific humidity measured by the Picarro laser instruments and the saturated vapour pressure at the surface

temperature (*Goff and Gratch*, 1945). Note that due to intake of snow crystals in the inlet of the Picarro, relative humidity is overestimated in very supersaturated conditions, but other hygrometers installed at Dome C confirmed supersaturated conditions with relative humidity ranging between 105% and 125% between 19:00 on the 6th of January and 06:00 on the 7th of January.

The evolution of water vapour and snow isotopic compositions is synchronous with observations of mist and solid condensation due to local large supersaturation. During this five hour period, water vapour $\delta^{18}O_v$ increased from -73‰ to -64‰ while snow $\delta^{18}O_s$ decreased by roughly 2‰. From 21:30 on the 6th of January (UTC time) to 01:40 on the 7th of January (UTC time), the volumes of three snow crystals were monitored by a script transferring the size in pixels of each crystals from the time-lapse sequence to surfaces using a length etalon and estimating the volume variations $\Delta V$ using a power law from the

surface variations $\Delta S$: $\Delta V \propto \Delta S^{3/2}$. This shows an increase by a factor from 1.5 to 3.9. The growth of the crystals observed in the time-lapse suggests a large-enough ice deposition to significantly affect the isotopic composition. The solid condensation occurs simultaneously with the modification of the isotopic composition of the snow and of the vapour. We observe a small delay (2 to 3 hours) between the beginning of the vapour isotopic composition increase and the decrease of the surface snow isotopic composition. These observations can be explained by an exchange of molecules between the snow and the vapour,

significantly affecting the snow isotopic composition and leading to an enrichment of the isotopic composition of the vapour and a depletion in the snow (note that the origin of the vapour can be either from the free atmosphere or from the pores in the snow; we are not able to discriminate between the two) .

### 3.3.2   Thermodynamics of the isotopic exchanges between the snow and atmosphere during a frost deposition in a closed box system

We evaluate the exchanges at the diurnal scale between the atmospheric vapour and surface snow using a simple thermodynamic model. This approach, similar to *Ritter et al.* (2016), is used here to simulate the depletion of heavy isotopes in the snow and the enrichment of the vapour. The system is composed of three boxes: (1) surface snow (roughly $1.5\,cm$ thick), (2) the atmospheric box representative of the atmospheric boundary layer able to exchange with the surface and where the vapour was sampled and (3) the free atmosphere considered as a homogeneous reservoir of water vapour. Compared to *Ritter et al.* (2016), this third box

provides a possible renewal of the air masses in the boundary layer. This conceptual model evaluates the expected variations of surface snow isotopic composition for the observed variations of water vapour isotopic composition during the frost deposition. The condensation flux at the snow surface is estimated from the bulk method around $1g.m^{-2}.h^{-1}$ (personal communication from E. Vignon, similar to *Genthon et al.* (2017)). For the period of 5 hours highlighted in Fig. 5, this corresponds to a transfer of $\Delta n_v = 0.3\,mol.m^{-2}$ of water molecules. Considering the variation of vapour pressure calculated from humidity monitoring





during this period ($\Delta P_v = 50 Pa$), this transfer requires the height of the atmospheric box in which the vapour is removed to be:

$$h = \frac{\Delta n_v}{\Delta P_v} RT = 12m \tag{1}$$

where $R$ is the molar gas constant and $T$ the air temperature. This height of $12\,m$ is in agreement with independent estimate of the boundary layer thickness considering the summer dynamic of the boundary layer at Dome C (*Vignon et al.*, 2017). The number of moles of $H_2^{18}O$ transferred to the snow surface, $\Delta n_{cond}^{18}$, is then:

$$\Delta n_{cond}^{18} = \underbrace{\Delta n_v^{18}}_{\substack{\text{Closed box like} \\ \text{contribution}}} + \underbrace{\Delta n_{ren}^{18}}_{\substack{\text{Air masses} \\ \text{renewal}}} \tag{2}$$

where $\Delta n_{cond}^{18}$ is the number of molecules condensed during the frost deposition, $\Delta n_{ren}^{18}$ is the number of molecules renewed in the atmospheric box by either advection, molecular or turbulent diffusion; and $\Delta n_v^{18}$ is the variation of heavy isotope molecules in the atmospheric box. Note that both the closed box contribution $\Delta n_v^{18}$ and the renewal $\Delta n_{ren}^{18}$ are here defined as positive contributions to the total amount of heavy isotopes condensing $\Delta n_{cond}^{18}$. In this framework, $\Delta n_v^{18}$ is estimated by:

$$\Delta n_v^{18} = R_v^{18} \Delta n_v + n_v \Delta R_v^{18} \tag{3}$$

Where $\Delta R_v^{18}$ is the variation of isotopic composition in the boundary layer. The contribution of heavy isotopes towards the surface snow in a closed box-like system is thus: $\Delta n_v^{18} = 5.6\ 10^{-4} mol.m^{-2}$. It is interesting to note that the contribution associated with the fractionation $n_v \Delta R_v^{18}$ accounts for less than 10% of the contribution of the closed box system.

The renewal of the atmospheric box with the free atmosphere can not be directly inferred here as we only measured the water vapour isotopic composition at $2\,m$ height. Based on summer isotopic composition profiles in Polar Regions (*Berkelhammer et al.*, 2016), we assume that the value of $R_v^{18}$ measured at a height of $2m$ is representative for the whole lower atmospheric box. We consider that during the 'day time', active turbulence has homogenised the vapour isotopic composition of the atmospheric column, and that during the 'night', due to more stable conditions, the isotopic composition above the atmospheric box has remained constant. Thus, in the particular case of a frost deposition, the term $\Delta n_{ren}^{18}$ is calculated as a fraction of $\Delta n_v^{18}$ by:

$$\Delta n_{ren}^{18} = \epsilon \left( n_v^{18} - n_v^{18}(init) \right) = \epsilon \Delta n_v^{18} \tag{4}$$

where $\epsilon$ is a parameter (varying between 0 and 1) which depends on advection and turbulent processes, and $n_v^{18}(init)$ is the initial number of heavy isotope set for the entire boundary layer before turbulence stopped. The quantity of $H_2^{18}O$ condensing





$\Delta n_{cond}^{18}$ condensing is thus defined as:

$$\Delta n_{cond}^{18} = \Delta n_v^{18} + \epsilon \Delta n_v^{18} \tag{5}$$

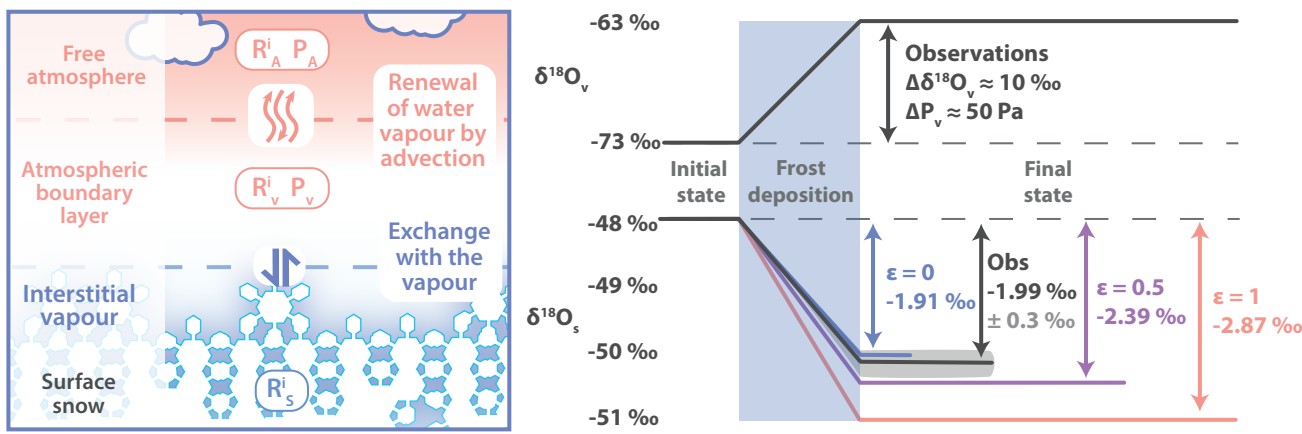

**Figure 6.** Schematic of the thermodynamic model for the exchanges between the snow and the vapour during condensation: the snow exchange with the vapour through equilibrium fractionation at the phase transition and the atmospheric vapour in the atmospheric boundary layer is only weakly renewed by advection of air masses (0 % : no advection, closed box system in blue; 50 % meaning that the contribution of the advection is equal to half the amount of water vapour involved in the phase transition (purple) and 100 % in orange.)

We apply this simple model to evaluate the variation of isotopic composition of the $1.5\,cm$ of surface snow associated with the condensation of $0.3\,mol.m^{-2}$ of water vapour for the cases of a closed box system ($\epsilon = 0\%$) or with the renewal of $100\%$ of the heavy isotopes (see Fig. 6). We find that for a closed box system ($\epsilon = 0$), the modelled variation of surface snow isotopic composition is 1.91 ‰ close to the observed value of $1.99 \pm 0.3$ ‰ in the surface snow $\delta^{18}O$ (see Fig. 5). In the case of a renewal of $100\%$ of the heavy isotopes in the atmospheric box ($\epsilon = 1$), we obtain a value of 2.87 ‰, which overestimates the

changes of surface snow isotopic composition.

This box model shows that at Dome C, the surface snow isotopic composition can be significantly affected by the deposition of frost in one example during one "night". At NEEM, *Steen-Larsen et al.* (2014) showed that the isotopic compositions of the snow and the vapour present a parallel increase in warm conditions in between precipitation events up to 7 ‰. Similar

observations have been made in Antarctica at Kohnen (*Ritter et al.*, 2016). However, at the diurnal scale, no parallel evolution is observed in snow and vapour isotopic compositions contrasting with results from NEEM and Kohnen (*Steen-Larsen et al.*, 2014; *Ritter et al.*, 2016). Our observations are consistent with exchanges in a closed system (renewal of up to 20 % of the





isotopes in the vapour): the vapour is enriched in heavy isotopes while snow is depleted during frost deposition events. We attribute the difference of behaviour compared to the parallel evolution observed at Kohnen (*Ritter et al.*, 2016) and NEEM (*Steen-Larsen et al.*, 2014) to the position of the station on the top of a dome. Indeed, at Dome C, the weakness of the katabatic winds ($3.3\ m.s^{-1}$) decreases the renewal of air masses long enough for the exchanges with the snow to be detected.

At Kohnen and NEEM, stronger winds are observed ($4.5$ and $4.1 m.s^{-1}$, respectively) leading to a more efficient renewal of air masses able to exchange with the surface (larger $\epsilon$ values). It is important to note that the humidity content at Dome C is lower than at Kohnen and NEEM, resulting in a smaller reservoir of water vapour with which the snow can exchange. Despite these low humidity levels, a significant impact of the sublimation/condensation cycles on the snow isotopic composition is observed. Similar studies measuring isotopic composition of vapour and snow at sites with similar temperatures but larger

wind speeds (such as at Vostok), could provide more robust insights on the impact of wind on the renewal of air masses compared to humidity levels. This study relies on one event of attested solid condensation and the monitoring of more events is necessary to be able to quantitatively evaluate the fractionation processes involved, however because of the asymmetry of the dynamic of the atmospheric boundary layer between night and day (stable at night during condensation, turbulent during the day during sublimation), we expect a more important renewal of the air masses during the day, leading to an anomaly of

isotopic composition after several sublimation/condensation cycles.

### 3.3.3  Evidence for isotopic modifications linked to snow metamorphism

We investigate whether, at the seasonal scale, the exchanges of heavy isotopes during sublimation/condensation cycles contribute to the seasonal signal, and therefore, whether the surface snow isotopic composition variations are linked to exchanges with the atmospheric water vapour, independent of the precipitation input. To do so, we use the grain index as an indicator of

grain coarsening. Indeed, the grain coarsening in the surface snow is due to successive sublimation/condensation with the pore vapour.

Figure 3 presents the grain index estimated from satellite data (*Picard et al.*, 2012) to evaluate the impact of metamorphism in the surface snow. Periods of strong metamorphism identified during the summer are highlighted (blue shaded areas). We

observe a link between summer grain index highs and the amplitude of the seasonal variation of $\delta^{18}O_s$ of the surface snow: in 2011 and 2015, small cycles of $\delta^{18}O$ are associated with a large grain index starting to increase in December; whereas in 2014 (and partially in 2012), the large summer increase of $\delta^{18}O_s$ is associated with small summer increase of grain index, in this case delayed after mid-January. The same pattern is not observed for precipitation (Fig. 3) whose seasonal isotopic variations appear more regular and in phase with temperature.

*Picard et al.* (2012) interpret grain-index variations as follows: high grain index values in the summer can be attributed to intense metamorphism (*Picard et al.*, 2012; *Libois et al.*, 2015); the rise usually starts during the first week of December. Rapid falls of the grain index result from important precipitation events and the input of small snow grains from precipitation. For instance, during summer 2015, we observe significant variations of the surface snow isotopic composition while no precipita-



tion input was identified, as described in section 3.2. During this period, intense metamorphism was observed, as highlighted by the grain index rise in Fig. 3. The variations of roughly 8 ‰ observed in the surface snow isotopic composition are in phase with the temperature variations. This supports the hypotheses of an input from the exchanges with the vapour: indeed, the snow is porous at Dome C, and thus the exchanges with the vapour could affect the first centimetres globally. This is discussed

more thoroughly in section 3.4. Finally, the slow decrease during winter is explained by the accumulation of new small snow flakes by precipitation onto the coarse grains formed during the summer. Winter metamorphism is too slow to impact the snow structure.

Finally, we find two features of interest regarding the amplitude of the variations of $\delta^{18}O_s$ of the surface snow in Fig. 3.

First, the inter-annual variability of the summer surface snow isotopic composition seems directly related to the strength of the metamorphism as suggested by the negative correlation between the amplitude of the grain-index increase in summer and the maximum $\delta^{18}O_s$ ($R^2 = 0.54$). However, this correlation appears to be independent of the link between metamorphism and the amount of precipitation (not shown), but a larger dataset is required to validate these preliminary results. The summer increase of $\delta^{18}O_s$ seems to be very sensitive to the date at which the intense summer metamorphism starts. The large val-

ues of $\delta^{18}O_s$ observed in 2014 (and maybe for the year 2012, but the maximum of $\delta^{18}O_s$ was reached before the sampling started) are associated with a small and delayed increase of grain index (in both case, the main increase of grain index happens after the 15th of January, whereas for normal years, it starts the first week of December). This delayed start of the metamorphism enables the surface snow to retain the enriched summer isotopic composition of precipitation. Second, in winter, we observe a mixing of new precipitation (depleted compared to summer) on top of the already deposited snow as illustrated by

the slow decrease of the $\delta^{18}O_s$ throughout the winter. This is similar to what is observed for the grain index. By contrast, there is no apparent relationship between the isotopic composition of precipitation and the grain index from 2008 to 2011 (see Fig. 3).

### 3.4 Surface and sub-surface snow isotopic exchanges

During the summer 2013/14, regular sampling of surface (0 to $3\,cm$) and sub-surface (3 to $6\,cm$) snow were carried out at

Dome C. Once a day, two samples of snow at each level were taken along with specific surface area (SSA) measurements to evaluate the size of the grains and therefore, how much metamorphism has taken place (*Picard et al.*, 2016b). These data are presented in Fig. 7 with temperature and precipitation from reanalysis and in-situ observations of precipitation. We observe that overall, during these two summer months, the sub-surface isotopic composition is almost systematically lower than the surface one. This is found as well in most of the snow pits presented in Section 3.5.

From the end of November to the 15th of December, we observe similar values of surface and sub-surface snow isotopic composition. This is a period during which metamorphism has not started yet as indicated by the large values of SSA (Fig. 7). The surface snow isotopic composition is low (around -55‰) and the SSA is high which is typical of winter snow. From the 16th of December, we observe large differences between the surface and the sub-surface snow isotopic composition (up to 5‰



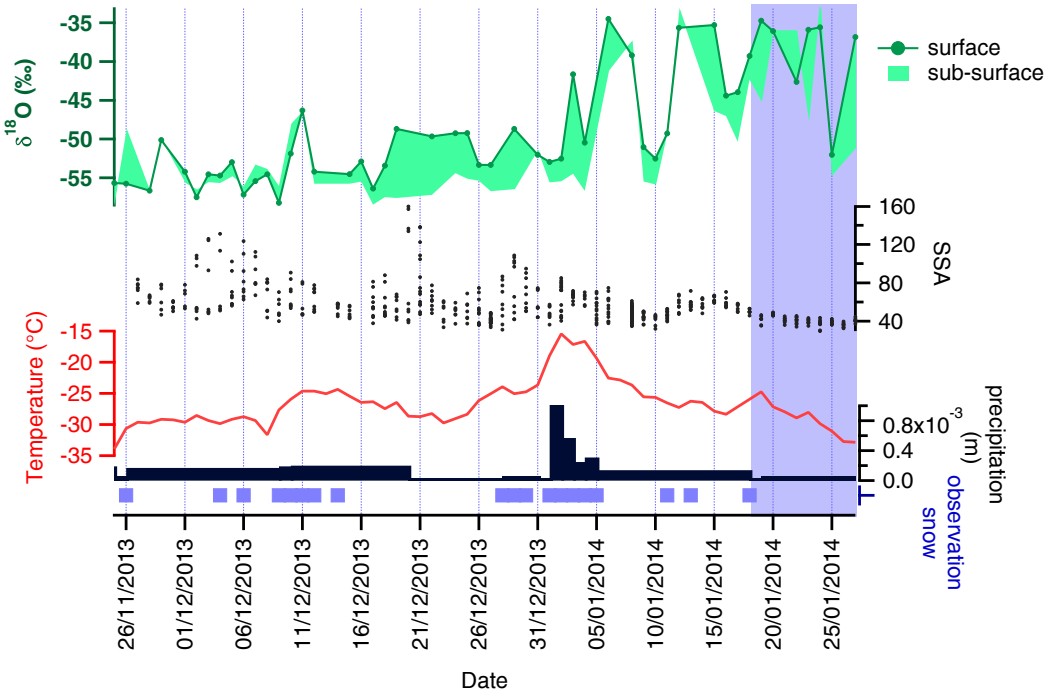

**Figure 7.** Isotopic composition of the snow at the surface (blue line) and difference between surface and the sub-surface (blue shade) during 2 months at Dome C in 2013/14, compared with SSA measurements (black dots), $2\,m$ temperature (red line) and precipitation (black bars) from ERA-interim, and precipitation observations (purple squares). The shaded area corresponds to the period during which the grain index changes reflect large metamorphism (see section 3.1.2).

higher at the surface) and the first decrease of SSA indicating the effects of metamorphism. This feature is not associated with significant precipitation input, and the difference between the surface and the sub-surface isotopic composition could be linked to post-deposition processes. Until the 31st of December, numerous drift events mix the snow and therefore cause strong spatial variability. Finally, due to a large precipitation event around the 2nd of January, we observe a significant increase

5 of snow $\delta^{18}O_s$ of 18 ‰ at the surface first and at the sub-surface no more than two days later. The increase of the sub-surface $\delta^{18}O$ cannot be explained only by the accumulation of snow as this event only accounts for roughly $1\,cm\,w.e.$ (which is already large compared to the annual accumulation of roughly $2.3\,cm\,w.e.$). Here, it is clear that the surface snow isotopic composition is directly affected by this precipitation event, whereas it seems that the subsurface snow isotopic composition is less sensitive to this impact and that it only changes as a delayed reaction to the surface changes.

As illustrated in Fig. 3, the grain index shows that strong metamorphism only starts in the middle of January (high grain index increase indicates strong metamorphism, in contrast, for surface SSA, accumulation of metamorphism results in a small SSA). During this period, we observe a large variability of both the surface and the sub-surface snow isotopic composition. It



is important to note here that the variations of isotopic composition include both spatial and temporal variations as only one sample per day was taken, therefore some of the variability might be due to spatial variability.

This sampling campaign suggests that: (1) snow metamorphism alters surface snow isotopic composition even in the absence of precipitation, and (2) precipitation isotopic composition can rapidly be transferred to the sub-surface. The most likely candidate for this signal transfer is molecular diffusion throughout the interstitial air, but more extensive time series will be necessary to quantify the processes involved.

### 3.5 Signal archived in the snow pack

We have observed that the surface snow isotopic composition signal is composed of several contributions, from the precipitation input to the exchanges of the atmospheric vapour. Both leave on a seasonal signal in the snow isotopic composition, which could be visible in vertical profiles of snow isotopic composition. We now focus deeper in the firn to evaluate whether the signal imprinted in the snow isotopic composition at the surface is preserved after the burial when accounting for isotopic diffusion as suggested by *Münch et al.* (2017) for Kohnen Station.

#### 3.5.1 Observation of apparent cycles

At Dome C, variations of $\delta^{18}O$ with depth are quite large within the top of the firn (typically of the order of 5 ‰) and irregular (Fig. 8). This feature has been confirmed by isotopic records on two $1\,m$ deep snow pits dug in 2014/15 at Dome C (Fig. 9). The two isotopic composition profiles, separated by roughly $100\,m$, are not well correlated ($R = 0.15$), as expected from stratigraphic noise (*Münch et al.*, 2016).

In order to evaluate whether these variations could be explained by intermittent precipitation events, and thus reflect climatic variability, we generated a profile of isotopic composition by accumulating snow using the relationship between precipitation isotopic composition and temperature, and the reanalysis products, as previously described in section 3.2 and in Supplementary Materials A. The results of this model include the precipitation intermittency and the temperature anomaly associated with large precipitation events. The results are presented in Fig. 9. In the synthetic snow isotopic composition profiles, we observe irregular cycles characterised by minimal values (winter) around -55 ‰ and maximal values (summer) either reaching up to -42 ‰, or limited to -50 ‰. The typical period of these cycles is around $8\,cm$, very close from the annual accumulation at Dome C ($7.7\,cm$).

The isotopic profiles obtained from the snow pits dug in 2014/15 do not show any comparable variability and especially no cycle with a periodicity close to the accumulation rate (around $8\,cm$). The spacing between $\delta^{18}O_N$ maxima in the profiles presented in both Fig. 8 and 9 present a systematic average value of $20\,cm$. As a result, if a periodical signal were used to depict the Dome C isotopic data, a $20\,cm$ cycle would be more appropriate than the expected $8\,cm$ annual cycles. However, no peak





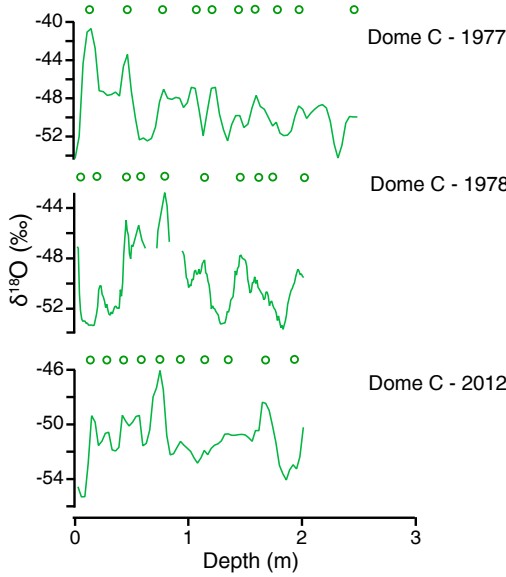

**Figure 8.** Isotopic composition profiles from 3 snow pits from Dome C, the successive maxima indicate the pseudo-cycles (circles) for each profile with a threshold of 1.5‰ in $\delta^{18}O_N$ between the successive local minima/maxima to prevent noise from artificially being counted as cycles.

is actually observed on the power spectra (*Laepple et al.*, 2017), so these $20\,cm$ variations can only be referred as 'apparent cycles'.

Thus, neither seasonal variability nor multi-year accumulation variations explain the $\delta^{18}O_N$ variability in snow pits observed

at Dome C.

### 3.5.2 Similarity of the apparent cycles across the East Antarctic Plateau

We extend the study of snow pit profiles found at Dome C to four other sites on the East Antarctic Plateau (Kohnen, S2, South Pole and Vostok) which are characterised by different meteorological and glaciological parameters such as mean annual temperature, elevation, wind speed and direction, accumulation or sastrugi height. A representative subsection of the profiles

of isotopic composition from the different sites is presented in Fig. 12 in Supplementary Materials B.

We evaluate the similarity of variability between the different sites by manual counting of the successive maxima. We again find a $20\,cm$ spacing on average between $\delta^{18}O_N$ maxima across sites of the Antarctic Plateau (Table 4, see Supplementary Materials B for details on the method). This signal is particularly robust for sites such as Vostok with seven snowpits with

'apparent cycle' lengths between 19 and $26\,cm$ and for Kohnen with 2 snowpits with 'apparent cycle' lengths between 17





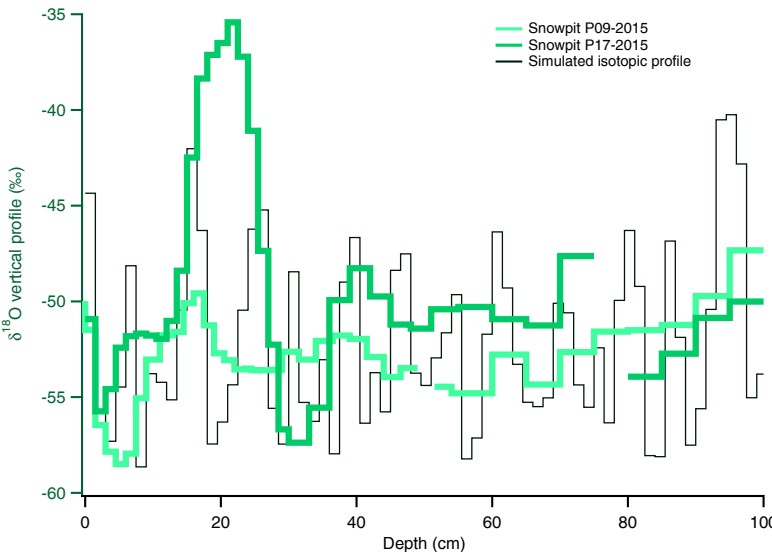

**Figure 9.** Isotopic composition profiles from two pits from Dome C (Green lines) retrieved in January 2015 compared to simulated isotopic composition profile from accumulation of precipitation (dark green thin line) and compared with the density and the temperature profile (black line and red dots, respectively).

**Table 4.** Mean 'apparent cycle' length obtained by manual counting of maxima from the isotopic composition profiles from the pits. Sites are sorted by accumulation in snow equivalent (calculated using an average snow density of $350\,kg.m^{-3}$).

| Site | Accumulation (cm of snow) | Spacing between max of $\delta^{18}O_N$ (cm) | Spacing between max of $\delta D_N$ (cm) | Number of pits | Length of the pits (m) | Finest Resolution (cm) |
|---|---|---|---|---|---|---|
| S2 | 6.0 | 24 | 20 | 1 | 2.6 | 3 |
| Vostok | 6.9 | 22 | 22 | 6 | From 2 to 12 | 2 |
| Dome C | 7.7 | 18 | 19 | 4 | From 1 to 3 | 1 |
| Kohnen | 18.3 | 19 | NA | 2 | 3 | 3 |
| South Pole | 19.7 | 20 | 20 | 2 | From 6 to 10 | 1.1 |

and $23\,cm$. Similar typical lengths are generally observed for the vertical profiles of snow $\delta D_N$ and $\delta^{18}O_N$, but our manual counting method, applied to a limited number of pits with relatively low resolution, would not enable to detect small differences.

For sites with high accumulation such as the South Pole and Kohnen (around $20\,cm$ of snow equivalent accumulation),
5    seasonal variability should be evident in snow isotopic composition variations (*Jouzel et al.*, 1983). In this case, the observed cycle lengths could simply reflect the imprint of seasonal variations in annual layers. Yet, the profiles are highly variable, exhibiting strong differences in between sites and as well as in between pits from a single site, even if sampled the same year. This can be attributed to the mixture of the potential climate signal and non-climate noise (*Fisher et al.*, 1985; *Münch*





*et al.*, 2016; *Laepple et al.*, 2016). Here, when we compare several snow pits dug the same year at the same sites (for instance at Kohnen as presented on Fig. 12 and observed on more profiles), we do not observe synchronous peaks in the profiles of isotopic compositions. Since multi-year cycles of the climatic conditions and thus isotopic composition of precipitation would globally affect one site, this finding suggests that the features we observe are the expression of non-climatic (post-deposition)

processes, resulting in spatial variability smoothed by diffusion (*Münch et al.*, 2017).

For sites with lower accumulation such as Vostok and S2, we observe similar results to those from Dome C: the spacing between $\delta^{18}O_N$ maxima are larger than expected from the annual accumulation rates. For these sites, in order to observe the seasonal variations (with an associated length of the order of the accumulation between 6 and $7\,cm$), a resolution finer than $3\,cm$ is necessary to resolve the seasonality (*Nyquist*, 1924; *Shannon*, 1949). The limited resolution of the S2 profile may thus

explain why no seasonal cycle of isotopic composition is visible. However, in the case of Vostok and Dome C, the vertical resolution of the isotopic composition profile is fine enough to establish the lack of seasonal cycle.

This indicates that non-climatic (post-deposition) processes are preponderant for the formation of these $\delta^{18}O_N$ variations in the firn. A forward model to evaluate how this signal is formed within the firn is out of the frame of this study and has been

thoroughly evaluated in the pair study from *Laepple et al.* (2017).

## 3.6 Relations between isotopic composition and temperature in the precipitation and in the snow

One major limitation in the isotopic paleothermometer is the uncertain and potentially variable relationship between isotopic composition and temperature. This is reflected in the range of slopes which are used to reconstruct temperature from isotopic composition including slopes obtained from precipitation at the seasonal scale, spatial slopes from samples covering several

years or temporal slopes from independent calibration at large temporal scale (higher than 100 years). Here, we investigate how the isotope-temperature slope is affected by the post-deposition processes described above, and if this could explain the different slopes found in the literature from different type of samples (precipitation, surface snow, buried snow).

First, we compare the relationship between the isotopic composition of precipitation and surface snow and temperature in

the model and the datasets. Figure 10a presents the isotopic composition - temperature relationship in the dataset of isotopic composition of precipitation and computed by the Mixed Cloud Isotopic Model (MCIM, see section 2.5). Except in summer (December, January, and February), the MCIM is able to faithfully simulate the isotopic composition of precipitation. The simulated relationship between $\delta^{18}O_p$ and temperature in the model shows a slope of $0.95‰°C^{-1}$ (see Table 5), similar to the one found from the data from the transect between Terra Nova Bay and Dome C which were used to tune the model (*Winkler*

*et al.*, 2012). For the entire seasonal cycle, we observe for the isotopic composition of precipitation a slope below $0.46‰°C^{-1}$. Strong differences are not unexpected between temporal and spatial slope of precipitation (*Ekaykin*, 2003; *Landais et al.*, 2012; *Touzeau et al.*, 2016). It is interesting to note here that the winter temporal slope of isotopic composition of precipitation ($0.76\,‰°C^{-1}$) matches the spatial slope of isotopic composition for the East Antarctic Plateau ($0.77\,‰°C^{-1}$ for low isotopic composition area) as illustrated in Fig. 10a. Here, the low slope of the entire seasonal cycle is due to the summer isotopic



composition which diverges from what is predicted by the MCIM (See the December-January-February datapoints in Fig. 10a) resulting in a slope of $0.41‰°C^{-1}$ for the entire year. One way to explain such a low slope would be to introduce additional fractionation linked with re-evaporation during the precipitation events which can affect the snow flakes' isotopic composition (*Koster et al.*, 1992) and therefore decrease the slope with temperature. This may also result from changes in air mass trajectory,

5  and therefore in the Rayleigh distillation. Backtrajectory calculations for the East Antarctic Plateau indicate strong asymmetry of the moisture sources for austral summer and winter (*Sodemann and Stohl*, 2009; *Winkler et al.*, 2012). Finally, in the MCIM, the condensation temperature is estimated through a linear relationship with the local surface temperature (*Ciais and Jouzel*, 1994). The reduced summer temperature inversion at Dome C (*Ricaud et al.*, 2014) is thus not taken into account in the MCIM which could also lead to a reduced slope.

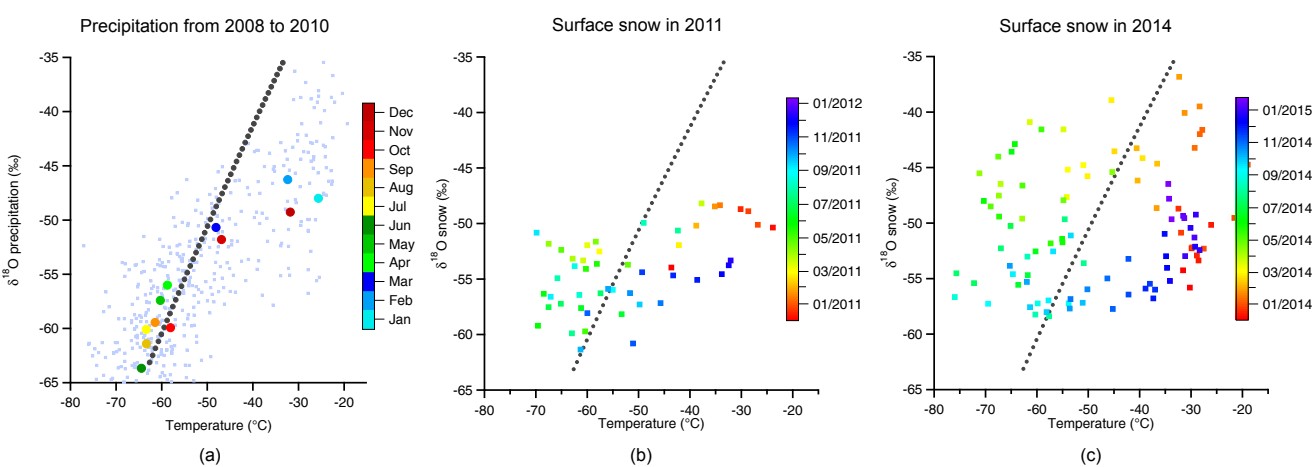

**Figure 10.** a) Isotopic composition of precipitation versus daily average of 3m temperature from 2008 to 2010 (light blue squares) and mean monthly values over these three years (coloured circles) from our datasets, compared with outputs of the MCIM for the same range of temperature tuned for Dome C (black dots); b) and c) respectively isotopic composition of surface snow in 2011 and 2014 versus daily average of 3m temperature.

This indicates that at least during winter, this model is able to predict the value of snow isotopic composition from mixed phase Rayleigh distillation with the tuning of supersaturation from *Winkler et al.* (2012) such as $S = 1 - 0.003T$, at least for winter conditions. This tuning of the supersaturation implies supersaturation up to $115\%$ in summer conditions at Concordia, consistent with the range of measurements obtained by *Genthon et al.* (2017).

By contrast, links between temperature and surface snow isotopic composition are not clear (Fig. 10). Thus, for Rayleigh type models to evaluate the isotopic signal in deposited snow, we expect that it is necessary to add a component taking into account 1. the integrator effect of the snow layer accumulation and 2. the post-deposition impacts. The range of values spanned





by the surface snow isotopic composition cycle is sometime coherent with the precipitation cycle (2014, see Fig. 10 c)) and sometimes not (2011 see Fig. 10 b); 2015, not shown). The difference of methods to sample the surface snow through the year does not explain this behaviour as for instance, the same protocol was applied to both 2014 and 2015. The surface snow isotopic composition is compared with the output of the MCIM model (Fig 10). These results confirm that in 2014, the snow

isotopic composition spans all the range predicted by the model for this range of temperature, as for the isotopic composition of precipitation; whereas for 2011 (and 2015, not shown), the surface snow isotopic composition does not rise above -45‰ and does not reflect the enriched summer values. The differences in the maximal summer values between 2014 and 2011 reported here are not linked to the isotopic composition of precipitation. Indeed, precipitation events with values of isotopic composition higher than -40 ‰ have been reported during all these summers and the isotopic composition of precipitation is in phase with

the temperature.

**Table 5.** Summary of the isotope temperature relationships observed for the different datasets. For the surface, because of the two-month shift, the slopes are calculated using the difference between the extrema of isotopic composition and temperature as detailed in section 3.1.2. For 2012, the summer maxima of isotopic composition were not sampled leading to an underestimation of the slope. For 2011, the dephasing is small enough to perform a linear regression; the result is indicated in parentheses. For the precipitation, the vapour and the MCIM output, we ran a linear regression. All the correlations are significant ($p-values < 0.05$). Summer is calculated over DJFM and winter over AMJJAS.

| Type of sample | Period | Slope $\delta^{18}O$ vs $T$ ($‰°C^{-1}$) | $r^2$ |
|---|---|---|---|
| Surface | 2011 | 0.22 (0.14) | 0.29 |
| | 2012 | $> 0.27$ | NA |
| | 2014 | 0.49 | NA |
| | 2015 | 0.27 | NA |
| Precipitation | All years | 0.46 | 0.65 |
| | Summer | 0.41 | 0.54 |
| | Winter | 0.76 | 0.56 |
| Vapour | Summer 2015 | 0.46 | 0.26 |
| MCIM | Multiyear | 0.95 | 0.99 |
| Transect to Dome C | Multiyear | 1.20 | 0.69 |
| | $\delta^{18}O$ < -40 ‰ | 0.77 | 0.90 |

Because the timeseries of surface snow $\delta^{18}O_s$ and of temperature are not in phase, it is not possible to directly estimate the corresponding temporal slope by linear regression. This is particularly important in 2014 when the amplitude of the isotopic composition cycle is greatest. We therefore estimate the relationship by comparing the peak to peak range in temperatures and isotopic composition. As the phase lag is smaller in 2011, we use that year to compare the peak-to-peak slope to the linear

regression. For 2011, during which the amplitude of the isotopic composition seasonal cycle is greatly reduced with respect to 2014, the slope between $\delta^{18}O_s$ and temperature is $0.14‰°C^{-1}$ (*Touzeau et al.*, 2016) rising up to $0.22‰°C^{-1}$ if considering the difference between maximum summer values and minimum winter values. For 2014, we obtain a slope of $0.49‰°C^{-1}$ ,





much lower than the prediction of the MCIM but closer to the value of the slope between precipitation $\delta^{18}O_p$ and temperature reported by *Touzeau et al.* (2016) of $0.46‰°C^{-1}$ at Dome C (see also *Stenni et al.* (2016) and Table 5).

In conclusion, the signal observed in the precipitation isotopic composition is already not entirely reflected in the first

centimetres of the surface snow. Averaging due to the sampling process and precipitation intermittency are both expected to impact the isotopic signal, however, our model suggests that they are not sufficient to explain all the difference in signal. Here, we identify that redistribution of the snow (for instance, by wind) can explain spatial and temporal short scale variations of surface snow isotopic composition but may not be sufficient to explain the signal obtained in the surface snow at the seasonal scale, and therefore prevents from using the slope of precipitation isotopic composition against temperature to reconstruct

climatic signal from ice records from Dome C at a seasonal scale.

## 4   Conclusions

In this study, we explored the post-deposition processes affecting the archiving of water isotopic composition from precipitation to the snow pack  focusing on Dome C, a low accumulation site on the East Antarctic Plateau.

First, we demonstrated that surface snow isotopic composition at Dome C is affected by post-deposition processes, in particular exchanges between the atmosphere and the snow pack, leading to a seasonal signal in the surface snow isotopic composition different from the one expected from the precipitation signal. The amplitude of this isotopic signal seems to be associated with the strength of the surface metamorphism. The post-deposition effect influences the relationship between $\delta^{18}O$ and temperature, which has important consequences for the interpretation of deep ice core water isotopes signal.

Second, we showed that at Dome C, similarly to other low-accumulation sites in East Antarctica, variations of the $\delta^{18}O$ signal with depth in shallow firn cores does not correspond to past climatic seasonal variations. The typical length associated with these variations or 'apparent cycles' is on the order of $20\,cm$, and differ for low-accumulation sites with the expected seasonal cycles.

Third, we illustrated how different can be the slope between surface snow isotopic composition and temperature compared to the slope between precipitation isotopic composition and temperature. We attributed this behaviour to the different fractionation and meteorological conditions involved in the different inputs that affect the surface snow isotopic composition between precipitation and various post-deposition processes.

Our study is a qualitative demonstration of the importance of post-deposition effects for isotopic signal at the surface and sub-surface of snow. A second step would be to address this process in a more quantitative way through controlled laboratory experiments, field studies and use of snow models equipped with water isotopes. The combined use of water isotopes ($d-$





*excess* or $^{17}O - excess$) may also be a strong added value due to their different relative sensitivity to equilibrium and kinetic fractionation.

*Acknowledgements.* The research leading to these results has received funding from the European Research Council under the European Union's Seventh Framework Programme (FP7/2007-2013) / RC grant agreement number 306045. T.M and T.L. were supported by the Initiative and Networking Fund of the Helmholtz Association Grant VG-NH900. We acknowledge the programs NIVO and GLACIO and all the IPEV staff that made the campaigns possible, LGGE and LIPHY for providing logistic advice and support, PNRA-PRE-REC project for the 2014 surface snow data at Concordia station.

## Appendix A: Simulation of the precipitation isotopic composition

The precipitation isotopic composition is simulated from the ERA-interim temperature and snowfall products and the relationship between precipitation isotopic composition and temperature obtained from section 3.1.1:

$$\delta^{18}O_p = 0.46 \times T - 32 \tag{A1}$$

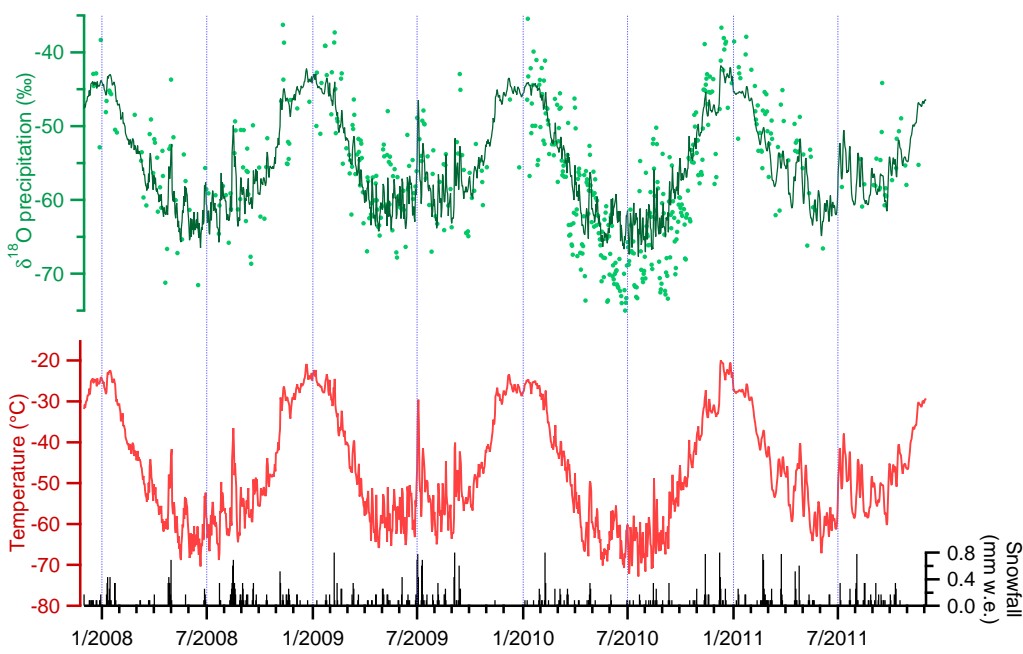

**Figure 11.** Comparison of isotopic composition of the precipitation (light green points) and simulated isotopic composition (dark green line) from the temperature (red) from ERA-interim



The simulated precipitation isotopic composition is compared to field measurements in Fig. 11: we observe that the simulated precipitation content matches the observations at the seasonal scale except for the very low values of isotopic composition observed in winter. This is striking in winter 2010 where the modelled $\delta^{18}O_p$ is capped down to -65 ‰ whereas values below -70 ‰ are frequency observed. The simulated seasonal cycle only captures 85 % of the observed seasonal cycle in terms of

amplitude.

At the day-to-day scale, high values of $\delta^{18}O_p$ associated with warm synoptic events are captured by the simulated precipitation product in winter. In summer, the modelled $\delta^{18}O_p$ fails to capture these warm events impact on $\delta^{18}O_p$.

Overall, the simulated isotopic composition signal captures most of the seasonal cycle observed in the datasets, in particular, it successfully models the variability level in winter, the temporality of the high $\delta^{18}O_p$ events and the lack of lag between precipitation isotopic composition and temperature.

**Appendix B: Signal in the snow pits across the East Antarctic Plateau**

We analyse the typical variations observed in the snow pit by manually counting the successive local extrema with a threshold

of minimum 1.5‰ for $\delta^{18}O_N$ and 10‰ for $\delta D_N$ for the difference between a minimum and a maximum (in both cases, the thresholds are chosen higher than the measurement precision and lower than the annual variations of surface snow isotopic composition; sensitivity tests have been carried out that show the impacts are not significant). For each snow pit, the mean cycle length is estimated by counting the number of maxima over the length of the pit. We present the average of the cycle length of the different pits for each site (Table 4).



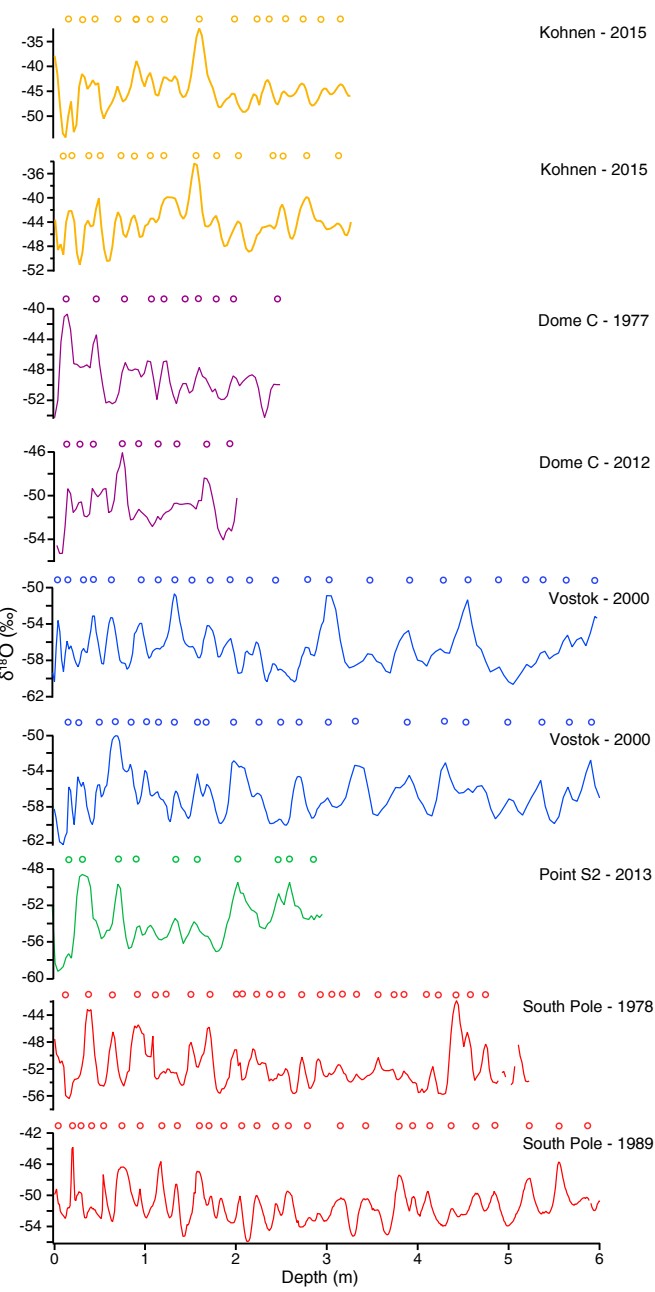

**Figure 12.** Isotopic composition profiles from 2 pits from Kohnen (Yellow), 2 pits from Dome C (Purple), 2 pits from Vostok (Blue), one pit from S2 (Green) and two pits from South pole (Red) and counting of cycles (circles) for each profile with a threshold of 1.5‰ in $\delta^{18}O_N$ between the successive local minima/maxima to prevent noise from artificially being counted as cycles.





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
