# Peer review of "Archival processes of the water stable isotope signal in East Antarctic ice cores"

_The Cryosphere, 2017_

## Referee Comment (RC1) · Anonymous Referee #1 · 22 Dec 2017

Review of „ Archival processes of the stable isotope signal in East Antarctic ice cores "
by Casado et al.

This paper presents a synthesis of various datasets of the isotopic composition of near-surface water vapour, precipitation, surface snow, buried snow from Dome C as well as snow pits from five other Antarctic sites. The analysis of this data focuses on gaining a better understanding of the processes that govern the temporal variability of the surface and buried snow isotope signals at the synoptic to seasonal timescales. The paper remains rather qualitative in the results presented and discussed, but a good overview of the relevant processes is provided. This includes a discussion on the input of fresh snow by snowfall, the influence of deposition-sublimation cycles, metamorphism, as well as redistribution by wind.

I read this paper with great interest, it provides a valuable overview and first analysis of important post-depositional processes affecting the water isotope signals at low-accumulation sites such as Dome C. However, I have several concerns of major and minor nature, which address a few methodological aspects as well as the description of the relevant processes.

Major Comments:
1) I found the discussion of the frost deposition event particularly interesting, because, I also believe that such deposition events during warm advection might play an important role for the snow cover isotope signal, particularly at low accumulation sites (Section 3.3.1). I was however puzzled by several aspects that need clarification. **First**, it is astonishing that during the period of frost deposition the air temperature $T_a$ is lower than the surface temperature $T_s$. I would expect it to be the other way around, which would hint towards an inversion layer near the ground that favours sensible and latent heat fluxes towards the surface. Is it really the case that $T_a<T_s$? If it is: Do the authors maybe have access to other sensors? The amplitude of $T_s$ is as large as the one of $T_a$, isn't this surprising for an event that the authors argue to be a synoptic warm advection case (I would expect a larger amplitude for $T_a$ than $T_s$, the snow temperature evolution being slower/dampened)? Was there any precipitation recorded during the event? The supersaturation is extremely large (Fig. 5). Is this the relative humidity with respect to a liquid or an ice surface?
**Second**, I am confused with the closed box model used in Section 3.3.2. Why don't you use isotope ratios instead of changes in number of molecules? How do you represent the phase change in this model? I don't find any fractionation factor relating the isotope ratio in the vapour to the one in the ice. Actually, if I am right, you seem to use the observed changes in $R18O$ in the vapour to infer the changes in the snow cover by mass conservation in a closed snow-vapour box. Could you make this clearer in the text? I suspect that there is a strong inversion (to be verified with the $T_a$-$T_s$ observations) which prevents strong mixing between the near-surface air and the air higher above in the boundary layer. This would be consistent with an epsilon value=0. Furthermore, if the closed box model assumption is good, you should be able to predict the time evolution of the vapour and snow phase isotope composition from a simple Rayleigh model with initial conditions from your observations around 18 UTC on the 6$^{th}$ of January 2015. Did you try this?

2) Statistical evaluation of the predictions from the precipitation and snow isotope models: I would find it very useful, if you could quantify the goodness of your simple precipitation and snow cover isotope model simulations by comparing them with your measurement data using scatter plots, mentioning the temporal correlation and the root mean square difference. This could be shown in the appendices A and B and the summary numbers could be mentioned in the main text.

3) The relevant processes discussed here are sometimes referred to in an unprecise manner. For example, is "sublimation-condensation cycles" really what you mean? Don't you mean frost deposition-sublimation cycles. It would be of great use for the reader if the 4-5 processes discussed as important in the paper were precisely defined in the introduction (which phase changes are meant?) and then reused consistently throughout the paper.

4) Something that disturbs me: why is the vapour isotope composition not more prominently mentioned for example by including the d18Ovapour signal in Figure 3? If I understood it correctly the main message of the paper is: "snow metamorphism matters, repeated frost deposition-freezing cycles alter the isotope composition of the snow". That would imply that the vapour isotope signature is mirrored into the snow, wouldn't it? Also in the perspective of other recent publications e.g. Steen-Larsen, et al. 2014 vapour-snow interactions seem to really play an important role in the "dry" time periods between precipitation events.

Minor/technical Comments:
1) Units should be in normal style, not italics
2) Abstract: p.1 L1-2: The first two sentences mention "records" (repetition). And I would add a sentence to link the oldest ice core records with the stable water isotope composition of the ice, before mentioning them as being important climate proxies of conditions over the ice and at the moisture source.
3) P. 1, L. 4: I would add "**trajectory-based** Rayleigh distillation and isotope-enabled climate models"
4) P.1, L. 7: I suggest "of **the** isotopic composition **of the snow later forming the ice core ice**"
5) P1, L. 8: "we combine observations of **the** isotopic composition"
6) P1, L. 11: I suggest "on **the isotopic signal of** the surface snow"
7) P1, I suggest to also mention the importance of the vapour isotope signal in between precipitation events in the abstract.
8) P2, L. 3: You could add Jouzel and Masson-Delmotte, 2010, **and references therein**
9) P2, L. 16: A philosophical question: I wondered whether these phenomena really all create **non-climate** signals? Could the strength of the spatial variability over longer timescale not be a measure for the importance and typical spatial scale of these post-depositional redistribution-related processes? And is the redistribution really homogeneous in time and not dependent on the local climate? For example, in periods of more frequent warm advection events and more stable stratifications couldn't the redistribution by wind be weaker? Or in other words: are local wind turbulence conditions not also somehow related to the frequency of different weather regimes and thus dependent on the climate?

10) P2, L. 19: to open more on your work, you could begin by: "**Thus an important open question that needs to be addressed is**, whether this seasonal cycle is archived or not…"

11) P2, L. 28: "between **the** isotopic composition"

12) P.2, L. 29: What do you mean by "boundary layer processes"? At the evaporative source or at the sink over the ice?

13) P.2, L. 31: Add in the "source" evaporation conditions

14) P.3, Fig. 1: In the caption indicate what Pv and Psat is (the partial pressure and the partial pressure at saturation?)

15) P. 4, L. 4: "Condensation" seems strange at such low temperatures. Is this really what you mean? So first a phase change from the vapour to the liquid phase and then freezing into the solid phase?

16) P. 4, L. 5: Why do you expect that? Could you argue a bit more explicitly? Is it because of the long inter precipitation-event duration and thus the long exposure of the surface snow to these processes?

17) P.4, L.11: Maybe add "net sublimation occurs" or "sublimation dominates over condensation".

18) P. 4, L. 14: "more **stably** stratified". Or probably even the build up of strong inversion layers. You mention "important temperature gradients observed". Can you say more about this? I.e. Do you observe inversion layers?

19) P.-4, L. 21: "acquired during **evaporation at the moisture source** and the formation of precipitation…"

20) P. 4, L. 32 "deliver and **discuss**"

21) P. 7, Table 2: I suggest "sampling rate or frequency in days" instead of "Resolution".

22) P. 8, L. 15: could you just mention the correlation coefficient and the root mean square difference between the measurement station data and ERA Interim at the 6-hourly and seasonal time scale? I don't expect ERA Interim to be too good at Dome C and it's the best estimate that you have, but just to know how good the reanalysis data is.

23) P.8, L12-18: it was not immediately clear for me why you write this paragraph. Please explicitly say this, i.e. mention that you use ERA-Interim for the modelled delta18O precipitation.

24) P.9, L1: you thus use the grain index as a proxy for the strength of metamorphism? Can you say that explicitly?

25) P.9, L.10: I somehow missed how you computed the air mass trajectory, could you please mention this?

26) P.9, section 2.5; I like your approach and organisation of the paper of trying first to explain the snow isotope signal by precipitation isotope input, and then introducing the toy precipitation-snow cover transfer model!

27) P. 9, L. 24: don't you mean specifically the exchanges with the **vapour phase** here? Exchange with the atmosphere would for me include the input of precipitation.

28) P. 9, L.23-26: this sentence is a bit long and the second part should be something like: "by relating the surface snow and precipitation isotopic composition to the meteorological conditions and the grain index".

29) P.10, section 3.1.1 I would find it very insightful to know what the average precipitation amount per event and the inter event precipitation duration is.

30) P.10, L. 13-15: what are the possible reasons for this difference in precip isotope-temperature slopes?
31) P. 12, L. 9: "these **warms** events are particularly visible in winter due to increased storminess in the sea ice margin in this season (Papritz et al. 2014, http://journals.ametsoc.org/doi/abs/10.1175/JCLI-D-13-00409.1)
32) P. 13, L. 17: "as well as the warm anomal**ies**"
33) P. 13, L. 22: "snow **water** equivalent"
34) P. 13, L. 25-26 it would be nice to know how well the ERA-Interim snowfall matches observations at the site (see also my major comment 2)
35) P. 14, L. 5: add a space between permil and are recorded
36) P. 14, L. 17: "interaction with the surface roughness" sounds a bit awkward and unprecise, what do you mean exactly?
37) P. 15, L. 4: "turbulent and convective atmospheric boundary layer" this surprises me, I would expect a very stable or even inversion thermodynamic layering in the boundary layer during such a warm event if indeed it was one. A synoptic map with the pressure reduced to sea level and the 500 hPa temperature distribution would maybe help to assess the large-scale weather situation.
38) P. 19, L. 8: "sublimation/condensation cycles" I would say frost deposition/sublimation cycles (see my major comment 3)
39) P. 20, L. 4: I am not convinced that "globally" is a good wording here.
40) P. 20, L. 12: "However,…." Rephrase this sentence, I got lost here.
41) P. 22, L. 5: "precipitation isotopic composition" is it? Don't you mean the surface snow isotopic signal? (see again my major comment 3)
42) P. 26, L. 13-14: remove the "at least for winter conditions at the end of the sentence. You already say it at the beginning of it.
43) P. 28, L. 15: "that **the** surface snow"
44) P. 28, L. 17: " The amplitude of **the snow** isotopic composition"
45) P. 28, L. 26: Reformulate the first sentence.
46) P. 28, L. 31: "for isotopic signal**s**"

All in all, this is an interesting interdisciplinary paper with innovative ideas!

---

## Referee Comment (RC2) · Anonymous Referee #2 · 4 Jan 2018

**Archival processes of the water stable isotope signal in East Antarctic ice cores.**

Mathieu Casado et al.

General comments: The submitted manuscript presents field measurements of water stable isotope signal in East Antarctic ice cores. Combine observations of isotopic composition in the vapor, the precipitation, the surface snow and the buried snow from Dome C were done. The results of this study are interesting: Surface snow isotopic composition is affected by post-deposition processes, in particular, exchanges between the atmosphere and the snow pack which were also observed in the laboratory work by Ebner et al. (2017). Further, the variations of the $\delta^{18}O$ signal with depth in shallow firn cores do not correspond to past climatic seasonal variations.

Specific comments: It is not easy to understand the manuscript. Especially the section with the results lacks a comprehensible structure. In order to get a context, the individual contents have to be gathered together over the sections. Additionally, more explanations of the extracted values are needed and some statements are far out because no work can be cited (only personal communication of researchers). In addition, important data (e.g. the depth where the samples were taken, …) are too much spread out throughout the whole manuscript, I would recommend adding these values into a table to have an easier overview. The nomenclature of the parameters in the text and figures are sometimes different and also some figures need better captions. Also, the comparison with other locations in the Antarctic is hard to understand as they have totally different conditions and sampling procedure. Finally, I would recommend plotting the extracted slopes to the corresponding data to see how well they fit.
Based on my comments I would suggest that the authors revise the manuscript carefully and rewrite the method and result part to make it better understandable for the reader. In my opinion, either a clearer focus is needed or the reasons for using the additional data sets must be clearly defined on an individual basis. Otherwise, I would recommend the editor to reject this paper. Although these data are very interesting, the authors are unable to explain and link them plausibly. For a publication of the manuscript in its present state, the structure is simply inadequate and some statements are formulated too vaguely. Beside good results, it is necessary to present it to the readers in a clear and comprehensible way.

Detailed comments:

Page 4, Line 16 – 22: There are experimental results where an interaction of the stable isotopes between snow and the surroundings were observed. More information can be found here: https://www.the-cryosphere.net/11/1733/2017/tc-11-1733-2017.pdf

Page 5, Line 7: "This study mainly focuses on Dome C,…" -> already 2 sentence later, you're explaining the comparison to all the others sites. So what are you really focusing on?

Page 5, Line 11: What do you want to say with the expression "joining"?

Page 6, Table 1: Why is the "AWS mean temperature" for Dome C equal to "NA" but they measured a "firn temperature" of -55.1?

Page 6, Line 13: What is a "clean area"

Page 6, Line 15: "average value of two samples" -> Can you say anything about the standard deviation?

Page 7, Line 4: "If the amount of snow on this second table was sufficient, …" -> What does "sufficient" mean?

Page 7, Line 7: "… that the protocol of surface snow sampling from the PRE-REC campaign differs greatly from the protocols from the NIVO and SUNITEDC programs due to the presence of the wood plate" -> If they "differs greatly", does it make sense to compare each other?

Page 7, Table 2: What does "Resolution" mean? Did they take for every e.g. 7 days samples? And I would also suggest mentioning the depth and sickness of the samples.

Page 7, Line 13: "For one of them, …"-> It would be good to mention which one was taken.

Page 7, Line 14: "plastic flasks" -> Were they air-tight and what did you analyze?

Page 7, Line 15: "… we compare the isotopic profiles to other snow pit samplings performed …" -> Where were the other snow pit samplings taken in East Antarctica? Do you have an overview on a map? Are they comparable?

Page 7, Line 20: "… beyond the decorrelation scale of the stratigraphic noise …" -> What is the distance to be "beyond the decorrelation scale of the stratigraphic" noise?

Page 8, Table 3: I would suggest adding more information, like the depth of the snow pits.

Page 8, Line 4: "… any impact of the sampling technique …" -> Why don't you expect any impact of the sampling and impact on what? Please provide some information.

Page 8, Line 15: "… we found a good agreement at the seasonal scale and fairly good agreement at the event scale (not shown here)." -> What does this statement mean for the presented results? And what is an "event scale"? Why don't you want to show it?

Page 9, Line 2: "When available, we include SSA measurements …" -> Why was it not possible to always include the SSA measurements?

Page 9, Line15: "… and the temporal slope of the isotopic …" -> What is the temporal slope? Can you provide some information?

Page 10, Line 27: "$\delta^{18}O_s$" -> Does this stand for the isotopic composition of snow? And is it the same like $\delta^{18}O_{snow}$ in Figure 3?

Page 11, Table 3: What is the resolution of these data, hourly, daily, averaged over the time, …?

Page 11, Line 1: "… are in agreement with the isotopic composition of precipitation (Dreossi, personal communication)" -> Where can I see it that it is in agreement? Please, can you provide the data of the personal communication, etc.?

Page 13, Line 13: "$\delta^{18}O_p$" -> Does this stand for the isotopic composition of precipitation? And is it the same as $\delta^{18}O_{precipitation}$ in Figure 3?

Page 13, Line 26: "The results of this modelled surface snow isotopic composition are …" -> Where is the modeled surface snow isotopic composition descripted?

Page 14, Line 3: "The model accurately reproduces some of the differences between the signal in the surface snow and in the precipitation …" -> It is hard to compare the signal in the surface snow and in the precipitation because there is only one year of overlapping (Figure 3).

Page 14, Line 9: "$\delta^{18}O_m$" -> Does this stand for isotopic composition of the model? And if yes, where is the model defined?

Page 15, Figure 5: Why are there some data missing (12:00 – 16:00) for the "Snow $\delta^{18}O$"? In the text, it is mention that every hour samples were taken.

Page 15, Figure 5: At which height have you measured the water vapor?

Page 15, Line 1: "… represents the noise on the surface snow due to the spatial variability" -> Have you extracted the noise from two measurements (Page 14, Line 32: "… two samples were taken from a random location …")?

Page 15, Line 4: Why is the exchange of moisture "important"?

Page 15, Line 7: "$\delta^{18}O_v$" -> Does this stand for the isotopic composition of vapor? And is it the same as "Vapour $\delta^{18}O$" in Figure 5? (uniform nomenclature?)

Page 16, Line 1: "… without being impacted by meteorological events …" -> Could you give more details: what do you mean by "meteorological events"?

Page 16, Line 7: "… ranging between 105% and 125% …" -> That is a big difference compared to "100% and 180%" shown in Figure 5. So, with the absorbed ice crystals the humidity graph in Fig 5. is worthless? Why don't you show the effective humidity measured by this "other hygrometers"? What other effects can the absorbed ice crystals have on the measured signal?

Page 16, Line 10: "… is synchronous with observations of mist and solid condensation due to local large supersaturation." -> Where have you observed the mist and solid condensation? Do you have any data?

Page 16, Line 32: "… (personal communication from …)" -> I see this explanation a bit questionable, please provide more details.

Page 16, Line 32 and 34 and following Lines: Remove the points between the units

Page 17, Line 14: "… $\Delta n_v^{18} = 5.6 \ 10^{-4}$ mol m$^{-2}$" -> How do you get this number? What value did you take for $R_v^{18}$, $\Delta n_v$, $n_v$, $\Delta R_v^{18}$? And what is $R_v^{18}$ and $n_v$?

Page 17, Line 15: "… with the fractionation $n_v \Delta R_v^{18}$ accounts for less than 10% …" -> Can you please provide more information? I cannot see how you get the 10%.

Page 18, Line 12: "… 1.91 ‰ close to the observed value of 1.99 ± 0.3 ‰ in the surface snow $\delta^{18}O$ (see Fig. 5)" -> Can you provide more information how do you extract the value of 1.99 ± 0.3 ‰ from Fig. 5? What is the initial and final state of the frost deposition?

Page 19, Line 1: "… the vapor is enriched in heavy isotopes while snow is depleted during frost deposition events." -> Is there an explanation for this? Why isn't it the opposite: If there is an exchange between vapor and snow, the vapor should be depleted in heavy isotopes because due to the higher mass than light isotopes, the heavy isotopes prefer more the solid state than the vapor state.

Page 19, Line 1-9: How can you compare your results at Dome C with other stations like Kohnen or NEEM if they have totally different conditions?

Page 19, Line 9: "Similar studies measuring …" -> Can you name them?

Page 19, Line 34: … during summer 2015, we observe significant variations of the surface snow isotopic composition while no precipitation input was identified, …" -> Please provide more information because according to Fig. 3 there was precipitation (snowfall) during summer 2015.

Page 20, Line 2: "… variations of roughly 8 ‰ observed in the surface snow isotopic composition are in phase with the temperature variations." -> I wouldn't say that they are in phase e.g. 1/2013 they are not in phase. I would recommend to say that there is a similarity in the variation.

Page 20, Line 16: "… are associated with a small and delayed increase of grain index (in both case, the main increase of grain index happens after the 15$^{th}$ of January, whereas for normal years, it starts the first week of December)." -> But what about 2011, it is also small and delayed and the increase is also at beginning of January.

Page 20, Line 21: "By contrast, there is no apparent relationship between the isotopic composition of precipitation and the grain index from 2008 to 2011." -> Please provide more explanation for this statement because the peaks between the isotopic composition of precipitation and the grain index matches well.

Page 21, Figure 7: The colors in the plot and in the caption are different.

Page 21, Line 1: "From the 16$^{th}$ of December, we observe … a first decrease of SSA indicating …" -> Due to the large variation in the SSA it is quite hard to say that there is a SSA decrease and how do you explain the high SSA around 20$^{th}$ of December?

Page 21, Line 3: "… numerous drift events mix the snow and therefore cause strong spatial variability." -> Do you have evidence for this conclusion?

Page 21, Line 4 – 14: -> Please provide more measurements/results to validate this statement. Why is there a sudden drop around 8$^{th}$ of January?

Page 22, Line 1: "… include both spatial and temporal variations as only one sample per day was taken, therefore some of the variability might be due to spatial variability." -> Does it make sense to use the data if you can have a variation of up to 18 ‰ which is quite large?

Page 22, Line 28: "… from the annual accumulation at Dome C (7.7 cm)." -> How did you get this value (7.7 cm)?

Page 22, Line 31: "… spacing between $\delta^{18}O_N$ …" -> What does "N" stand for?

Page 22, Line 31: "…between $\delta^{18}O_N$ maxima in the profiles … present a systematic average value of 20 cm" -> I cannot see systematic maxima in these graphs but a variation between 20 cm and 40 cm …

Page 23, Figure 8: At which depth were the snow pits taken? How did you make sure that the snow samples were air-tight, especially from the year 1977 and 1978?

Page 23, Line 14: "… Vostok with seven snowpits with …" -> It's six according to Table 4.

Page 24, Line 1: "… but our manual counting method, applied to a limited number of pits with relatively low resolution, would not enable to detect small differences." -> What do you mean by "small differences"? What differences?

Page 24, Line 8: "… of the potential climate signal and non-climate noise." -> What do you mean by "non-climate noise"? Is it a local signal?

Page 25, Line 10: "The limited resolution of the S2 profile may thus explain why no seasonal cycle of isotopic composition is visible." -> Please mention again the resolution of the S2 profile. In this statement, you say that no cycle is visible of the S2 profile but in Figure 12 a cycle of isotopic composition is visible…

Page 25, Line 28: "… similar to the one found from the data from the transect between Terra Nova Bay and Dome C …" -> Please provide the number.

Page 25, Line 26 – Page 26, Line 2: Please show in Figure 10 all the extracted slope you mention in this section.

Page 26, Line 8: "The reduced summer temperature inversion at Dome C is thus not taken into account in the MCIM which could also lead to a reduced slope." -> Does it

make sense to compare the Model with measurements? How big is the reduced summer temperature inversion?

Page 27, Line 15: "As the phase lag is smaller in 2011 …" -> Which "phase lag"?

Page 27, Line 14 – Page 28, Line 2: Please show the extracted slopes in Figure 10.

Page 29, Line 1: "(d – *excess* or $^{17}O$ – *excess*)" -> I would recommend to change it to "($d_{excess}$ or $^{17}O_{excess}$)"

---

## Author Comment (AC1) · 22 Mar 2018

Dear editor,

We would like to thank the two reviewers for the constructive and detailed discussion. Please find below our answers to the individual comments. The questions are repeated in black, while our answers are in blue and the suggested modifications of the manuscript in red.

On the behalf of all the co-authors,

Mathieu Casado

**Anonymous Referee #1**

This paper presents a synthesis of various datasets of the isotopic composition of near-surface water vapour, precipitation, surface snow, buried snow from Dome C as well as snow pits from five other Antarctic sites. The analysis of this data focuses on gaining a better understanding of the processes that govern the temporal variability of the surface and buried snow isotope signals at the synoptic to seasonal timescales. The paper remains rather qualitative in the results presented and discussed, but a good overview of the relevant processes is provided. This includes a discussion on the input of fresh snow by snowfall, the influence of deposition-sublimation cycles, metamorphism, as well as redistribution by wind.

I read this paper with great interest, it provides a valuable overview and first analysis of important post-depositional processes affecting the water isotope signals at low- accumulation sites such as Dome C. However, I have several concerns of major and minor nature, which address a few methodological aspects as well as the description of the relevant processes.

Major Comments:

1) I found the discussion of the frost deposition event particularly interesting, because, I also believe that such deposition events during warm advection might play an important role for the snow cover isotope signal, particularly at low accumulation sites (Section 3.3.1). I was however puzzled by several aspects that need clarification. **First**, it is astonishing that during the period of frost deposition the air temperature Ta is lower than the surface temperature Ts. I would expect it to be the other way around, which would hint towards an inversion layer near the ground that favours sensible and latent heat fluxes towards the surface. Is it really the case that Ta<Ts? If it is: Do the authors maybe have access to other sensors? The amplitude of Ts is as large as the one of Ta, isn't this surprising for an event that the authors argue to be a synoptic warm advection case (I would expect a larger amplitude for Ta than Ts, the snow temperature evolution being slower/dampened)? Was there any precipitation recorded during the event? The supersaturation is extremely large (Fig. 5). Is this the relative humidity with respect to a liquid or an ice surface?

We thank the reviewer for raising this very interesting question. It is indeed not intuitive. Such effect could be explained by the following hypothesis.

First, we observe that the surface temperature cycle is as large as the air temperature (even larger for most days, see in Casado et al. (2016)). This is due to radiative forcing, indeed, even though there is no real night, the modulation of the solar influx creates really strong variations of the surface snow temperature. The surface snow is then itself forcing the air temperature variations. This is the classical daily cycle description. The synoptic event here is preventing to study after 6:00 on the 7$^{th}$ of January, but we don't believe that it has influenced the warming phase prior to the condensation event.

Then, to why is it condensing when the surface temperature is warmer than the air temperature and not before. We observe that the atmosphere is oversaturated from the beginning of the night with very high supersaturation starting around 18:00 UTC. Yet, at this point, the time lapse video does not show any expansion of the crystals, even though, the surface temperature is lower than the atmosphere temperature, and thus, thermodynamically speaking, condensation should be observed. There, at this point, there is a blocking parameters preventing condensation from happening, even though thermodynamic conditions allow it. Considering the inversion layer, we believe that at this point, the lower layers of the atmospheric boundary layers are stratified (no observation of turbulence is available for this period though), and thus, the only process enabling transfer of molecules from the atmosphere to the surface is diffusion, which is rather inefficient (Scheme a), in figure R1).

On the other hand, after 18:00, we observe simultaneously that 1. the surface temperature get higher than the atmosphere temperature and 2. that frost deposition is observed at the surface. Our hypothesis (still, without

any observation of turbulence) is that the warmer temperature at the surface triggered convection in the lower boundary layer, enable turbulent exchanges (Scheme b), in figure R1). At this point, the lower atmospheric boundary layer is still oversaturated against the snow surface, therefore, the enhanced turbulent exchanges provide a much larger flux of water molecule at the surface than diffusion.

This stops after 01:00 as the temperature has increased and the lower boundary layer is not supersaturated against the surface, even though, the turbulent exchanges are still active (Scheme c), in figure R1).

[Figure]

Figure R1: Modified from the manuscript Figure 5, with the addition of schematics to describe the exchanges between the surface and the atmosphere.

Unfortunately, this mechanism cannot be proved yet, in the absence of the appropriate data, and we see this as future prospective work. This is the reason why we prefer not to include this discussion in the manuscript yet.

During this event, no precipitation was observed. Nevertheless, mist was observed.
The relative humidity is calculated relatively to the ice saturated vapour pressure.

2) **Second**, I am confused with the closed box model used in Section 3.3.2. Why don't you use isotope ratios instead of changes in number of molecules? How do you represent the phase change in this model? I don't find any fractionation factor relating the isotope ratio in the vapour to the one in the ice. Actually, if I am right, you seem to use the observed changes in R18O in the vapour to infer the changes in the snow cover by mass conservation in a closed snow-vapour box. Could you make this clearer in the text? I suspect that there is a strong inversion (to be verified with the Ta-Ts observations) which prevents strong mixing between the near-surface air and the air higher above in the boundary layer. This would be consistent with an epsilon value=0. Furthermore, if the closed box model assumption is good, you should be able to predict the time evolution of the vapour and

snow phase isotope composition from a simple Rayleigh model with initial conditions from your observations around 18 UTC on the 6$^{th}$ of January 2015. Did you try this?

We have implemented a significant amount of modifications to section 3.3.2 to clarify the section, and included most of the suggestions. First, we are using number of molecules to simplify the equations. The use of ratios would be also correct, but this would require complex differential calculation as both the amount of $H_2^{18}O$ and $H_2^{16}O$ are simultaneously changing.

Indeed, we are not including any isotopic fractionation and only implemented mass conservation in the model. This has been precised in section 3.3.2. This approximation seems realistic as the variation of the isotopes in the vapour only provides a small contribution to the total number of heavy isotopes which are actually condensing. The phase change is thus only represented by the transfer of molecules (calculated through the mass conservation) and no a priori values from fractionation coefficients are used. In a perfect world, both approaches should give the same results, but here, it is not the case. This has been included in the manuscript (page 18, line 12):

> "Yet, our observations and modelling results appear to disagree with this approach, mainly due to the fact that the fractionation coefficients are only able to describe thermodynamic equilibrium conditions, which were not met in our case study. "

Finally, the model we have developed is indeed a "semi-open" Rayleigh model, and in theory one could predict time evolution of both the vapour and the snow isotopic composition as suggested. Here, the major point actually preventing this calculation is that we only rely on estimates of the amount of vapour condensing, and do not have precise measurements to evaluate the precise mass balance. More precise mass balance estimates would require 3 D air masses movements and humidity measurements to be able to compute a complete mass balance.

3) Statistical evaluation of the predictions from the precipitation and snow isotope models: I would find it very useful, if you could quantify the goodness of your simple precipitation and snow cover isotope model simulations by comparing them with your measurement data using scatter plots, mentioning the temporal correlation and the root mean square difference. This could be shown in the appendices A and B and the summary numbers could be mentioned in the main text.

We have included statistical evaluations of the precipitation and snow isotopic models against observations. These calculations are indeed very useful and provide good evaluation of how much of the variance we can explain with our models. The evaluations have been included in the appendices. We have produced the scatter plots, but they don't provide any additional information compared to the temporal correlation and the root mean square difference. Considering the large number of figures, we would rather keep them out, but are open to send them to the reviewers. We have included discussions about these new results in section 3.2, and Appendix A.

4) The relevant processes discussed here are sometimes referred to in an unprecise manner. For example, is "sublimation-condensation cycles" really what you mean? Don't you mean frost deposition-sublimation cycles. It would be of great use for the reader if the 4-5 processes discussed as important in the paper were precisely defined in the introduction (which phase changes are meant?) and then reused consistently throughout the paper.

We included recaps of the definition in the introduction, and linked them with the schematic, as suggested by the reviewer

> "We chose Dome C as an open air laboratory to study the different contributions to the surface snow isotopic composition, including : (1) direct precipitation input, (2) blowing snow, (3) exchanges with atmospheric vapour and (4) exchanges with the firn below the surface (Fig. 1). Point (3) includes both sublimation and condensation (both liquid and solid condensation). The term "deposition" will refer to the deposition of precipitation and rime at the snow surface leading to accumulation. Point (4) includes several processes such as sublimation in warmer areas of the firn, molecular diffusion in the porosity sometimes enhanced by wind pumping, and solid condensation in colder areas of the firn. Point (4) can also be associated with metamorphism (coarsening of the snow grains as a result of temperature gradients in the firn), in which case the impact on the isotopic signal is similar to "isotopic diffusion" such as described by Johnsen (1977). Throughout the manuscript, the notations

used in Fig. 1 will be used to describe the isotopic composition of which type of snow is described."

5) Something that disturbs me: why is the vapour isotope composition not more prominently mentioned for example by including the d18Ovapour signal in Figure 3? If I understood it correctly the main message of the paper is: "snow metamorphism matters, repeated frost deposition-freezing cycles alter the isotope composition of the snow". That would imply that the vapour isotope signature is mirrored into the snow, wouldn't it? Also in the perspective of other recent publications e.g. Steen-Larsen, et al. 2014 vapour-snow interactions seem to really play an important role in the "dry" time periods between precipitation events.

We would actually think it would make sense to be able to include the vapour isotopic monitoring on Figure 3, unfortunately, only a few months (1 in 2014/15 and 1.5 in 2015/16) are available, because, to our knowledge, there isn't any instrument able to measure the vapour isotopic composition in winter in such a site (humidity below 1 ppmv according to (Genthon et al., 2013). Because the vapour isotopic composition will not mirror what is presented here for the snow (Only 2 months of monitoring of the vapour against several year-long time series for the snow), we would rather not include the vapour isotopic composition time series for the time being as the figure gets even more complicated to read.

Our present study provides qualitative results about the archival processes of the isotopic signature at Dome C, but we agree with the reviewer that to be able to quantify the transfer function of the exchanges between the snow and the vapour on the isotopic signal, we would require year-long vapour measurements.

Minor/technical Comments:

1) Units should be in normal style, not italics

We don't really know why this is the case, it seems to be some issue with the latex package. This has been corrected.

2) Abstract: p.1 L1-2: The first two sentences mention "records" (repetition). And I would add a sentence to link the oldest ice core records with the stable water isotope composition of the ice, before mentioning them as being important climate proxies of conditions over the ice and at the moisture source.

The repetition has been removed. We agree that it is important to link the stable isotopes to the oldest ice records; this has been included in the beginning of the introduction. In the abstract, we would rather not extend too much the length.

3) P. 1, L. 4: I would add "**trajectory-based** Rayleigh distillation and isotope-enabled climate models"

Included

4) P.1, L. 7: I suggest "of **the** isotopic composition **of the snow later forming the ice core ice**"

We understand the point of the reviewer here. Yet, here, we believe that it's the ice core isotopic composition interpretation which will be limited. We tried to modify the sentence to reflect your suggestion:

"In low accumulation sites, such as those found in Antarctica, these poorly constrained processes are likely to play a significant role and limit the interpretability of an ice core's isotopic composition.."

5) P1, L. 8: "we combine observations of **the** isotopic composition"

Included

6) P1, L. 11: I suggest "on **the isotopic signal of** the surface snow"

Included

7) P1, I suggest to also mention the importance of the vapour isotope signal in between precipitation events in the abstract.

Included

"Overall, we observe in between precipitation events modification of the surface snow isotopic composition"

8) P2, L. 3: You could add Jouzel and Masson-Delmotte, 2010, **and references therein**

Included

9) P2, L. 16: A philosophical question: I wondered whether these phenomena really all create **non-climate** signals? Could the strength of the spatial variability over longer timescale not be a measure for the importance and typical spatial scale of these post-depositional redistribution-related processes? And is the redistribution really homogeneous in time and not dependent on the local climate? For example, in periods of more frequent warm advection events and more stable stratifications couldn't the redistribution by wind be weaker? Or in other words: are local wind turbulence conditions not also somehow related to the frequency of different weather regimes and thus dependent on the climate?

Well, this is a very good remark. It is not the primary climate signal we are aiming for. Indeed, using several isotopic compositions and removing the signal due to the temperature, one could theoretically retrieve climatic information from these post-depositional redistribution related processes from the ice core records. Nevertheless, considering the uncertainty so far about the temperature dependency of the isotopic signal, this is purely conceptual.

We have included a more precised description in the main text (Page 2, line 16):

"Such phenomena create a degree of noise that is unrelated to past climatic conditions, and which could be alleviated by stacking different isotopic composition profiles from several snow pits to reveal the underlying climatic signal"

10) P2, L. 19: to open more on your work, you could begin by: "**Thus an important open question that needs to be addressed is**, whether this seasonal cycle is archived or not…"

Included

11) P2, L. 28: "between **the** isotopic composition"

Modified

12) P.2, L. 29: What do you mean by "boundary layer processes"? At the evaporative source or at the sink over the ice?

We meant locally. This has been precised.

13) P.2, L. 31: Add in the "source" evaporation conditions

Included.

14) P.3, Fig. 1: In the caption indicate what Pv and Psat is (the partial pressure and the partial pressure at saturation?)

Included.

15) P. 4, L. 4: "Condensation" seems strange at such low temperatures. Is this really what you mean? So first a phase change from the vapour to the liquid phase and then freezing into the solid phase?

The phase transition from Vapour to Solid is also referred as *"Condensation",* or *"Solid Condensation"*, the term here was directly taken from Genthon et al, 2017:

*"The flux is positive during the summer months indicating sublimation of snow, while during winter months the flux is negative, indicating condensation to the surface."*

I think this comes from discrepancies in the vocabulary between the different communities. We include a note (Page 3, line 13):

"associated with condensation (vapour-ice phase transitions with or without a liquid intermediary)"

16) P. 4, L. 5: Why do you expect that? Could you argue a bit more explicitly? Is it because of the long inter precipitation-event duration and thus the long exposure of the surface snow to these processes?

Precisions about this assertion have been included (Page 3, line 25):

"while this may appear small, we expect a significant impact on the snow isotopic budget: the alternation of negative and positive fluxes would overall have a small mass budget (symmetric mass balance), as the temperature is lower during the Austral winter when positive fluxes are observed than during the Austral summer when the negative fluxes are observed, the isotopic budget is affected by different isotopic fractionation for positive/negative fluxes periods

(asymmetric mass balance)..”

**17)** P.4, L.11: Maybe add "net sublimation occurs" or "sublimation dominates over condensation".

Even though, it is true that during the summer month, we overall have a net sublimation, here, the point is really to say that the temperature is different during the sublimation phase than during the condensation phase. We have modified this phrase for clarity (Page 3, line 33):

"As the sublimation phase (daytime) is characterised with higher temperature than the condensation phase ("nighttime")"

**18)** P. 4, L. 14: "more **stably** stratified". Or probably even the build up of strong inversion layers. You mention "important temperature gradients observed". Can you say more about this? I.e. Do you observe inversion layers?

A strong inversion layer has been observed during summer month in January 2015, the impact of this on the vapour isotopic composition has been described in (Casado et al., 2016). Overall, such situations are more frequent during winter, as described in (Vignon et al., 2017). Thus, we observe stratified conditions. We expect that they will reduce the amount of sublimation/condensation compared to turbulent conditions. For the isotopic impact, it is not clear as these stratified conditions are linked to more important kinetic processes which are only poorly studied (would be the equivalent of (Merlivat and Jouzel, 1979) for snow/ice).

**19)** P.-4, L. 21: "acquired during **evaporation at the moisture source** and the formation of precipitation…"

Included

**20)** P. 4, L. 32 "deliver and **discuss**"

Included

**21)** P. 7, Table 2: I suggest "sampling rate or frequency in days" instead of "Resolution".

Modified

**22)** P. 8, L. 15: could you just mention the correlation coefficient and the root mean square difference between the measurement station data and ERA Interim at the 6- hourly and seasonal time scale? I don't expect ERA Interim to be too good at Dome C and it's the best estimate that you have, but just to know how good the reanalysis data is.

This is a very good suggestion, it has been included. We are nonetheless rather surprised by how "good" ERA Interim actually is, considering the location. (Page 8, line 20)

"we found a good agreement at the seasonal scale and fairly good agreement at the event scale ($R^2$=0.89, the mean difference is 6.1°C, the root mean square difference is 4.8°C)."

**23)** P.8, L12-18: it was not immediately clear for me why you write this paragraph.
Please explicitly say this, i.e. mention that you use ERA-Interim for the modelled delta18O precipitation.

Included (page 8, line 20):

"For this reason, all the modelling efforts realised in this manuscript use ERA-interim data in order to provide a consistent quality of data through the different period."

**24)** P.9, L1: you thus use the grain index as a proxy for the strength of metamorphism?
Can you say that explicitly?

Included

**25)** P.9, L.10: I somehow missed how you computed the air mass trajectory, could you please mention this?

In this model, we don't compute the air masses trajectory. It is a Rayleigh Distillation model that is tuned (quantity of water vapour remaining in the clouds at each condensation step, co-existence of liquid and solid phases, quantity of re-evaporation, …) using the transect snow isotopic composition. Only the starting and ending point are prescribed. Then the simulated isotopic data for a temperature gradient along a Rayleigh distillation were compared to isotopic data for comparable temperature range.

**26)** P.9, section 2.5; I like your approach and organisation of the paper of trying first to explain the snow isotope signal by precipitation isotope input, and then introducing the toy precipitation-snow cover transfer model!

Thanks a lot.

**27)** P. 9, L. 24: don't you mean specifically the exchanges with the **vapour phase** here?
Exchange with the atmosphere would for me include the input of precipitation.

Sentence deleted during the review.

28) P. 9, L.23-26: this sentence is a bit long and the second part should be something like: "by relating the surface snow and precipitation isotopic composition to the meteorological conditions and the grain index".

Sentence deleted during the review.

29) P.10, section 3.1.1 I would find it very insightful to know what the average precipitation amount per event and the inter event precipitation duration is.

This has been tried subsequently to the suggestion. The main issue is that because the only precipitation evaluation we have is ERA-interim and that the obtained values are heavily biased by the model, we are not sure how trustworthy are the results. The average precipitation event (when precipitation is observed, thus strictly above 0mm) is 0.19 mm. This occurs roughly 51% of the days, and the average period without precipitation is 1.75 days.

30) P.10, L. 13-15: what are the possible reasons for this difference in precip isotope- temperature slopes?

This has been precised (Page 9, line 27):

"The rather wide range of slopes between precipitation isotopic composition and temperature is due to different source regions, distillation paths and local conditions such as the temperature inversion (Landais et al., 2012; Winkler et al.,2012) and more details are provided in Section 3.6"

31) P. 12, L. 9: "these **warms** events are particularly visible in winter due to increased storminess in the sea ice margin in this season (Papritz et al. 2014, http://journals.ametsoc.org/doi/abs/10.1175/JCLI-D-13-00409.1)

Included.

32) P. 13, L. 17: "as well as the warm anomal**ies**"

Included

33) P. 13, L. 22: "snow **water** equivalent"

Here, we actually mean that it is not in water equivalent as classically given, but in snow amounts taking into account that the density of snow is variable for each site and lower than the water density.

34) P. 13, L. 25-26 it would be nice to know how well the ERA-Interim snowfall matches observations at the site (see also my major comment 2)

We could do such a comparison for the temperature averages, it is much more complicated to realise such a comparison at Dome C considering the lack of any kind of measurements of precipitation. Indeed, due to very cold temperature, frost formation and very low amount of precipitation, there is no reliable precipitation amount measurement. The closest measurements are local accumulation measurements, which is what we used to correct ERA-interim precipitation amounts using (Genthon et al., 2015).

35) P. 14, L. 5: add a space between permil and are recorded

Corrected

36) P. 14, L. 17: "interaction with the surface roughness" sounds a bit awkward and unprecise, what do you mean exactly?

Sentence deleted during the review.

37) P. 15, L. 4: "turbulent and convective atmospheric boundary layer" this surprises me, I would expect a very stable or even inversion thermodynamic layering in the boundary layer during such a warm event if indeed it was one. A synoptic map with the pressure reduced to sea level and the 500 hPa temperature distribution would maybe help to assess the large-scale weather situation.

Out of the East Antarctic Plateau, this kind of conditions would be associated with large-scale weather situation indeed. Here, at Dome C, in between any kind of synoptic events, we often observe during the summer convective and turbulent conditions during the day time. This has been studied in depth by several papers including Vignon et al, 2017 a) and b). We have included those two articles into this presentation of the site.

38) P. 19, L. 8: "sublimation/condensation cycles" I would say frost deposition/sublimation cycles (see my major comment 3)

We agree that this terminology matter is a major issue. We hope that the comment in the beginning of the manuscript about this will solve this problem.

39) P. 20, L. 4: I am not convinced that "globally" is a good wording here.

This has been changed for (Page 20, line 9):

> "which means that exchanges with the vapour could affect the upper centimetres simultaneously."

40) P. 20, L. 12: "However,…." Rephrase this sentence, I got lost here.

This has been changed to (Page 20, line 16):

> "Nevertheless, as there is a direct link between metamorphism and precipitation amount (Picard et al., 2012), the correlation between the summer surface snow isotopic composition and metamorphism could be coincidental. More samples in combination with reliable precipitation estimates are required to validate these preliminary results."

41) P. 22, L. 5: "precipitation isotopic composition" is it? Don't you mean the surface snow isotopic signal? (see again my major comment 3)

Modified

42) P. 26, L. 13-14: remove the "at least for winter conditions at the end of the sentence.
You already say it at the beginning of it.

Modified

43) P. 28, L. 15: "that **the** surface snow"

Included

44) P. 28, L. 17: " The amplitude of **the snow** isotopic composition"

Modified

45) P. 28, L. 26: Reformulate the first sentence.

Modified

46) P. 28, L. 31: "for isotopic signal**s**"

Modified

All in all, this is an interesting interdisciplinary paper with innovative ideas!

We would like to thank the reviewer 1 for these constructive comments which we believe improved the quality of the paper.

**Anonymous Referee #2**

General comments: The submitted manuscript presents field measurements of water stable isotope signal in East Antarctic ice cores. Combine observations of isotopic composition in the vapor, the precipitation, the surface snow and the buried snow from Dome C were done. The results of this study are interesting: Surface snow isotopic composition is affected by post-deposition processes, in particular, exchanges between the atmosphere and the snow pack which were also observed in the laboratory work by Ebner et al. (2017). Further, the variations of the $d^{18}O$ signal with depth in shallow firn cores do not correspond to past climatic seasonal variations.

Specific comments: It is not easy to understand the manuscript. Especially the section with the results lacks a comprehensible structure.

The structure of the manuscript has been under substantial modifications. First, we have added in the introduction a paragraph refining the terminology we are using throughout the paper:

"We chose Dome C as an open air laboratory to study the different contributions to the surface snow isotopic composition, including : (1) direct precipitation input, (2) blowing snow, (3) exchanges with atmospheric vapour and (4) exchanges with the firn below the surface (Fig. 1). Point (3) includes both sublimation and condensation (both liquid and solid condensation). The term "deposition" will refer to the deposition of precipitation and rime at the snow surface leading to accumulation. Point (4) includes several processes such as sublimation in warmer areas of the firn, molecular diffusion in the porosity sometimes enhanced by wind pumping, and solid condensation in colder areas of the firn. Point (4) can also be associated with metamorphism (coarsening of the snow grains as a result of temperature gradients in the firn), in which case the impact on the isotopic signal is similar to "isotopic diffusion" such as described by Johnsen (1977). Throughout the manuscript, the notations used in Fig. 1 will be used to describe the isotopic composition of which type of snow is described"

The Material and methods section has been focused and shorten, including only the necessary information and implementing links to original studies for already published dataset. The different tables in the section 2 have been completed with more of the relevant information. Systematic references to these different tables have been implemented throughout the manuscript in order to facilitate the comparison of the different protocols used to gather these data. Additionally, in the Figure 3 has been included a visual aid to evaluate which isotopic data was realised in which campaign.

Section 3.2 has been split with two subsections to highlight on one hand the Toy model description, and on the other hand, the comparison with the observations. The discussions have been pushed at the end of the section. A more extensive description of the toy model has been implemented in Supplementary material A.

Section 3.3.2. has been deeply rewritten in order to facilitate the understanding, and sub-sections have been introduced and the order of the calculation have been changed to separate on one hand water mass balance out of isotopic considerations, and then on the other hands, the isotopic calculations. Finally, the discussion has been separated from the results. Section 3.3.3 has been mainly rewritten.
Section 3.4 as well.

A global effort of separating the discussion from the results in all sub-section of section 3 has been realised. But we have mainly kept discussions and results in section 3, as considering the large amount of presented materials, we believed it's easier to go through the manuscript to have the results discussed as soon as presented. Overall, the manuscript has been shorten of roughly 1 page.

In order to get a context, the individual contents have to be gathered together over the sections. Additionally, more explanations of the extracted values are needed and some statements are far

out because no work can be cited (only personal communication of researchers). In addition, important data (e.g. the depth where the samples were taken, …) are too much spread out throughout the whole manuscript, I would recommend adding these values into a table to have an easier overview.

Information are included in the different tables in section 2. We have kept several tables separated in order to have one for each type of dataset, especially because the relevant information are not the same for snow pits or surface snow for instance. A more systematic referencing to these tables has been implemented throughout the manuscript to facilitate the overview of the work performed.

The nomenclature of the parameters in the text and figures are sometimes different and also some figures need better captions. Also, the comparison with other locations in the Antarctic is hard to understand as they have totally different conditions and sampling procedure. Finally, I would recommend plotting the extracted slopes to the corresponding data to see how well they fit.

Based on my comments I would suggest that the authors revise the manuscript carefully and rewrite the method and result part to make it better understandable for the reader. In my opinion, either a clearer focus is needed or the reasons for using the additional data sets must be clearly defined on an individual basis. Otherwise, I would recommend the editor to reject this paper. Although these data are very interesting, the authors are unable to explain and link them plausibly. For a publication of the manuscript in its present state, the structure is simply inadequate and some statements are formulated too vaguely. Beside good results, it is necessary to present it to the readers in a clear and comprehensible way.

A global effort of introducing the comparison to other sites more precisely has been realised. We believe these comparisons are accurate as even though temperature conditions may be different, the studies were usually targeting the exact same processes than we did. As describe above, a global distinction between discussion and results have been done and the structure has been modified.

Detailed comments:

Page 4, Line 16 – 22: There are experimental results where an interaction of the stable isotopes between snow and the surroundings were observed. More information can be found here: https://www.the-cryosphere.net/11/1733/2017/tc-11-1733-2017.pdf

The study has been included in the manuscript. It is indeed perfectly relevant.

Page 5, Line 7: "This study mainly focuses on Dome C,…" -> already 2 sentence later, you're explaining the comparison to all the others sites. So what are you really focusing on?

We are focusing on Dome C. Most of the presented results are about Dome C. Because we want to compare our results to similar type of results obtained in Antarctica, we recall the different kind of conditions observed in these stations. We have included an introduction to this comparison (Page 6, line 1):

"To provide a context for the situation of Dome C, we present the conditions found at Dome C compared to other deep ice core sites on the East Antarctic Plateau:"

Page 5, Line 11: What do you want to say with the expression "joining"?

This has been replaced (Page 6, line 2).
"campaign Explore-Vanish between Dome C and Vostok"

Page 6, Table 1: Why is the "AWS mean temperature" for Dome C equal to "NA" but they measured a

"firn temperature" of -55.1?

The AWS mean temperature equal to "NA" is for S2. Firn temperature and AWS (Automated Weather Station) are two different types of measurement. No AWS was installed at S2, so there are no measurements available.

    Page 6, Line 13: What is a "clean area"

The clean area is a standard term to describe an area kept clean by preventing any type of pollution or dirt to contaminate a site. In Antarctica, it refers to the areas where the vehicles are not allowed to go, and upwind from station generators. For simplicity, we removed this reference as all samples were taken in a "clean area".

    Page 6, Line 15: "average value of two samples" -> Can you say anything about the standard deviation?

This information and conclusions about this are already included in section 3.1.2. A link to the section will be added. We want to keep the results together in the result section (Here, section 3), and not mix them up with the presentation of the sampling section.

    Page 7, Line 4: "If the amount of snow on this second table was sufficient, …" -> What does "sufficient" mean?

We have included an approximate value of the minimum amount of snow required for the sampling. Here, the measurement requires less snow than the sampling itself, thus, it is really limited by the ability of sampling small amounts. The sentence has been removed during the review.

    Page 7, Line 7: "… that the protocol of surface snow sampling from the PRE-REC campaign differs greatly from the protocols from the NIVO and SUNITEDC programs due to the presence of the wood plate" -> If they "differs greatly", does it make sense to compare each other?

It makes sense to compare both methods at least in the perspective of future surface sampling and we thought that it can be useful for some readers to know that several methods for surface sampling are used. Precisions about the difference have been included, and a link to the result section where the comparison of these two methods is done (Page 7, line 8):

    "The use of a wooden surface limits mixing with the snow below (both mechanical mixing of snow layers and diffusion/metamorphism) and represents the main difference compared to the other sampling methods. The comparison of the two sampling methods is provided in section 3.1.2."

    Page 7, Table 2: What does "Resolution" mean? Did they take for every e.g. 7 days samples? And I would also suggest mentioning the depth and sickness of the samples.

"Resolution" has been replaced by "Sampling rate". The depth and the thickness are included by "precipitation", "surface" and "sub-surface". A description in the caption has been added for the depth and the thickness meant by "surface" and "sub-surface" (Page 6, line 11)

    "Precipitation samples were collected at 1 a.m. when precipitation occurred. Surface snow refers to the top 5 to 30 mm of the firn, and sub-surface snow refers to depths between 30 and 60 mm."

Additional information concerning the different sampling campaigns have been included in the table so it can be used as a summary for the reader throughout the manuscript.

    Page 7, Line 13: "For one of them, …"-> It would be good to mention which one was taken.

Included (Page 7, line 16):

    "For the snow pit P09-2015, snow temperature and density profiles were established."

    Page 7, Line 14: "plastic flasks" -> Were they air-tight and what did you analyze?

They were air-tight indeed, and water isotopes solely were analysed. The sentence has been updated (page 7, line 16):

    "Snow samples for isotopic analysis were taken in airtight plastic flasks."

    Page 7, Line 15: "… we compare the isotopic profiles to other snow pit samplings performed …" -> Where were the other snow pit samplings taken in East Antarctica? Do you have an overview on a map? Are they comparable?

The link to the map has been included at this location too.

Page 7, Line 20: "… beyond the decorrelation scale of the stratigraphic noise …" -> What is the distance to be "beyond the decorrelation scale of the stratigraphic" noise?

The distance has been included with an additional reference to the Munch 2016 paper.

Page 8, Table 3: I would suggest adding more information, like the depth of the snow pits.

The depths have been included in Table 3.

Page 8, Line 4: "… any impact of the sampling technique …" -> Why don't you expect any impact of the sampling and impact on what? Please provide some information.

This sentence has been removed from this section as it is not strictly speaking data presentation.

Page 8, Line 15: "… we found a good agreement at the seasonal scale and fairly good agreement at the event scale (not shown here)." -> What does this statement mean for the presented results? And what is an "event scale"? Why don't you want to show it?

The "event scale" means that we look at the comparison between ERA interim product and local temperature at the time scale of several days, typically the scale of synoptic events. A detailed comparison between ERA interim product and local temperatures throughout Antarctica is an important task which should be done in great details and would be a paper by itself. Still, we have included statistical analysis to better assess this comparison (Page 8, line 20):

"we found a good agreement at the seasonal scale and fairly good agreement at the event scale ($R^2$=0.89, the mean difference is 6.1°C, the root mean square difference is 4.8°C)."

Page 9, Line 2: "When available, we include SSA measurements …" -> Why was it not possible to always include the SSA measurements?

SSA was not always part of the programmed measurements realised on the field.

Page 9, Line15: "… and the temporal slope of the isotopic …" -> What is the temporal slope? Can you provide some information?

The scale has been precised closer to the assessment (Page 9, line 8):

"The model results provided a comparison between the spatial (estimated from observations from 10 to 1000 km apart) and the temporal relationships (estimated from seasonal variations) of the isotopic composition of precipitation and were used to quantify the impact of post-deposition processes by providing a reference for the precipitation isotopic composition."

Page 10, Line 27: "d$^{18}$O$_s$" -> Does this stand for the isotopic composition of snow? And is it the same like d$^{18}$O$_{snow}$ in Figure 3?

Yes, it has included in the caption of Figure 3, and also of Figure 1, and throughout the text.

Page 11, Table 3: What is the resolution of these data, hourly, daily, averaged over the time, …?

We assume you are mentioning Figure 3. All the resolutions were indicated in section 2.2. Additional links to this section have been included for clarification.

Page 11, Line 1: "… are in agreement with the isotopic composition of precipitation (Dreossi, personal communication)" -> Where can I see it that it is in agreement? Please, can you provide the data of the personal communication, etc.?

It is impossible to provide the data to general audience but we can share them with the reviewer. They haven't been published by our Italian collegues and they have a separated publication in preparation on this subject. Please find below a low quality caption (Figure R2). We hope that these results from Dreossi et al will be published shortly.

[Figure]

Figure R2: precipitation isotopic composition in 2014 compared to the surface snow. The axes have been removed and the image distorted to respect the wishes of the owners of the data.

Page 13, Line 13: "$d^{18}O_p$" -> Does this stand for the isotopic composition of precipitation? And is it the same as $d^{18}O_{precipitation}$ in Figure 3?

Yes, it has been included in the caption of Figure 3, and also of Figure 1, and throughout the text.

Page 13, Line 26: "The results of this modelled surface snow isotopic composition are
…" -> Where is the modeled surface snow isotopic composition descripted?

Additional introduction to the toy model has been included at the beginning of section 3.2. for clarity (Page 12, line 6)

"The large variability in the amount of snow deposited during each precipitation event can influence the surface snow d18Os and create a different signal from that observed in the precipitation. We implemented a toy model to create synthetic precipitation isotopic composition and evaluate if the accumulation of several precipitation events captures the surface snow isotopic signal.."

Additionally, subsections have been included.

Page 14, Line 3: "The model accurately reproduces some of the differences between the signal in the surface snow and in the precipitation …" -> It is hard to compare the signal in the surface snow and in the precipitation because there is only one year of overlapping (Figure 3).

A more precise description to guide the reader has been included.

Page 14, Line 9: "$d^{18}O_m$" -> Does this stand for isotopic composition of the model? And if yes, where is the model defined?

It has been precised.

Page 15, Figure 5: Why are there some data missing (12:00 – 16:00) for the "Snow
$d^{18}O$"? In the text, it is mention that every hour samples were taken.

The samples were taken from 16:00 UTC time (midnight local time). Then they were taken every hour. Nevertheless, we believe that to introduce other available measurements from 12:00 to 16:00 provides an interesting context to the dataset we present here.

Page 15, Figure 5: At which height have you measured the water vapor?

This has been included in the manuscript (Page 14, line 9).

"The vapour isotopic composition was monitored at a height of 2 m."

Page 15, Line 1: "… represents the noise on the surface snow due to the spatial variability" -> Have you extracted the noise from two measurements (Page 14, Line 32: "… two samples were taken from a random location …")?

We have extracted the noise using the three measurements: 2 samples taken from a random location and one sample taken at a fixed point. This has been precised in the manuscript (Page 14, line 10):

"The spatial variability of the surface snow isotopic composition was estimated from hourly triplicate sampling at one fixed and two random locations (chosen from within a 30 m$^2$ area) ."

Page 15, Line 4: Why is the exchange of moisture "important"?
Changed to "strong".

Page 15, Line 7: "d$^{18}$Ov" -> Does this stand for the isotopic composition of vapor? And is it the same as "Vapour d$^{18}$O" in Figure 5? (uniform nomenclature?)
This has been included in the Figure 5, and then throughout the text. We also added another time "vapour isotopic composition d18Ov" for clarity.

Page 16, Line 1: "… without being impacted by meteorological events …" -> Could you give more details: what do you mean by "meteorological events"?
We use "meteorological events" as a generic term which includes all type of synoptic events but also different sort of precipitation, rime, wind… Considering the large number of possible events affecting the system, we will not be able to be exhaustive if we list them all, thus the use of a generic term. We have nonetheless precised that we minimise their impact, and not work without them which would be impossible in a field experiment. (See Page 14, line 20)

"In order to be able to study the exchanges between snow and vapour while minimising the impact of meteorological events,"

Page 16, Line 7: "… ranging between 105% and 125% …" -> That is a big difference compared to "100% and 180%" shown in Figure 5. So, with the absorbed ice crystals the humidity graph in Fig 5. is worthless? Why don't you show the effective humidity measured by this "other hygrometers"? What other effects can the absorbed ice crystals have on the measured signal?
It is a very good question. We'd rather consider that this is a very good indication that condensation is occurring in the lower atmosphere, both on condensation nuclei and at the snow surface. Thus, we believe we can use it as a proxy of intense condensation.

On the other hand, to say that commercial hygrometers would provide effective humidity is an overstatement. There are a lot of evidence that these hygrometers underestimate the supersaturation in this type of conditions (Genthon et al., 2017).

Finally, it is a very interesting point that these absorbed ice crystals can have an impact on the measured signal. As described in Casado et al, 2016, we have filtered out sublimated crystals from our system. Obviously, below a certain amount of ice, it will not be possible to distinguish these crystals from the natural variations of vapour isotopic composition. On the other hand, the question of the number of particles nucleated from which you can consider that a nucleus is not part of the vapour phase anymore is still open to our knowledge.

A link to our previous paper in which some these questions were already discussed has been included (Page 14, line 25)

"More information can be found in Casado et al. (2016)".

Page 16, Line 10: "… is synchronous with observations of mist and solid condensation due to local large supersaturation." -> Where have you observed the mist and solid condensation? Do you have any data?
Yes, this is included in the article, and is described just after this introduction sentence… We included another time the link to the video presenting this, which is also part of the supplementary material (Page 14, line 27):
"The evolution of water vapour and snow isotopic compositions is coincident with observations of mist and solid condensation due to local large supersaturation as evidenced by visual observations (five hour period in the blue shaded area in Fig. 5, see the time-lapse video in supplementary material)"

Page 16, Line 32: "… (personal communication from …)" -> I see this explanation a bit questionable, please provide more details.
The details are provided in the quoted article, in which we have done exactly the same calculation and which

was reproduced here. We have moved this part to the acknowledgements.

Page 16, Line 32 and 34 and following Lines: Remove the points between the units

Included.

Page 17, Line 14: "… $Dn_V{}^{18} = 5.6\ 10^{-4}$ mol m$^{-2}$" -> How do you get this number? What value did you take for $R_V{}^{18}$, $Dn_V$, $n_V$, $DR_V{}^{18}$? And what is $R_V{}^{18}$ and $n_V$?

We have taken into account the values that are presented in the figure 5 and that we will provide openly at the time of the final publication. We have calculated $R_V{}^{18}$ from the vapour $d^{18}O_V$, all the variations from the changes of humidity transferred into number of molecules. We have included a more precise description from the provenance of the data used here and linked it to Figure 6 where all the numerical values are included in a schematic. More numerical values were also included (Throughout Page 16, and specifically on Page 17, line 8):

"For the case study from Section 3.3.1, given that we observe changes of water partial pressure of 50 Pa and of isotopic composition of roughly 10 ‰ (See Fig. 6 for more details) , the contribution of heavy isotopes towards the surface snow in a closed box-like system isDnv18 = 5.6 10^-4 mol m^-2. The contribution associated with the fractionation nvDR18v < 5 10^-5 mol m^-2 accounts for less than 10% of the contribution of the closed box system.

Page 17, Line 15: "… with the fractionation $n_V DR_V{}^{18}$ accounts for less than 10% …" -> Can you please provide more information? I cannot see how you get the 10%.

The numerical values have been included.

Page 18, Line 12: "… 1.91 ‰ close to the observed value of 1.99 ± 0.3 ‰ in the surface snow $d^{18}O$ (see Fig. 5)" -> Can you provide more information how do you extract the value of 1.99 ± 0.3 ‰ from Fig. 5? What is the initial and final state of the frost deposition?

The figure 6. has been updated to better present these initial and final state of the frost deposition.

Page 19, Line 1: "… the vapor is enriched in heavy isotopes while snow is depleted during frost deposition events." -> Is there an explanation for this? Why isn't it the opposite: If there is an exchange between vapor and snow, the vapor should be depleted in heavy isotopes because due to the higher mass than light isotopes, the heavy isotopes prefer more the solid state than the vapor state.

This kind of consideration only works in a context of a thermodynamic equilibrium, which we are clearly out of. Once the effect of diffusion on fractionation is included, the theory predicts that the vapour should be enriched in heavy isotopes. Here, we decided not to use the fractionation coefficients considering the large uncertainties at low temperature. The explanation is actually provided in this entire chapter, by using the mass balance (even though we work with number of molecules). We believe that this is a particularly interesting case, because it shows how outdated are the fractionation coefficients in the context of mass balance and mass budget. This has been precised in the manuscript (Page 18, line 8):

"The box model showed that the surface snow isotopic composition at Dome C can be significantly affected by the formation of frost which is surprising considering that the isotopic fractionation is typically interpreted using equilibrium fractionation coefficients. Based on this typical interpretation, the solid phase should get enriched with heavy isotopes while the vapour phase should become depleted. Yet, our observations and modelling results appear to disagree with this approach, mainly due to the fact that the fractionation coefficients are only able to describe thermodynamic equilibrium conditions, which were not met in our case study. "

Page 19, Line 1-9: How can you compare your results at Dome C with other stations like Kohnen or NEEM if they have totally different conditions?

Precisions have been given for the comparison. We believe it is important to check if the impact of post deposition (in particular here, the exchanges between the snow and the atmospheric vapour) was valid for other sites, and if yes, if it was in the same conditions. :

"To put this into a wider context, we present other parallel measurements of vapour and snow isotopic compositions in summer in Polar Regions. At NEEM, Steen-Larsen et al. (2014) showed that the isotopic compositions of the snow…"

Page 19, Line 9: "Similar studies measuring …" -> Can you name them?

This sentence has been moved to the discussion part of the section, this was a perspective.

Page 19, Line 34: … during summer 2015, we observe significant variations of the surface snow isotopic composition while no precipitation input was identified, …" -> Please provide more information because according to Fig. 3 there was precipitation (snowfall) during summer 2015.

In summer 2015, precipitation events were very sparse. Here, we describe the periods in between precipitation events where no precipitation input was identified. While there was no precipitation, we still observe snow surface isotopic composition, as described previously in section 3.2. This has been changed to be more specific (Page 20, line 5).

"During summer 2015, we observed significant variations in the surface snow isotopic composition during periods without precipitation input"

Page 20, Line 2: "… variations of roughly 8 ‰ observed in the surface snow isotopic composition are in phase with the temperature variations." -> I wouldn't say that they are in phase e.g. 1/2013 they are not in phase. I would recommend to say that there is a similarity in the variation.

This sentence was specifically about summer 2015. We have included a new reference closer to this sentence to the summer 2015 (Page 20, line 5):

"During summer 2015, we observed significant variations in the surface snow isotopic composition during periods without precipitation input (see above) and intense metamorphism as highlighted by the increase in grain index (Fig. 3). The variations of roughly 8 ‰ of d18Os are in phase with the temperature variations.."

Page 20, Line 16: "… are associated with a small and delayed increase of grain index (in both case, the main increase of grain index happens after the 15$^{th}$ of January, whereas for normal years, it starts the first week of December)." -> But what about 2011, it is also small and delayed and the increase is also at beginning of January.

In 2011, the grain index increase is very **large** and starts end of December/beginning of January, which is different from the small increase observed in 2012 and 2014 where we have small increases (half the amplitude of 2011) and delayed (2 to 4 weeks, which is more than half of the summer duration). We agree with the reviewer that more events than 2 for both situations would be necessary to describe a strict relation. We added a reference to the casual relation that is observed here in the text, as well as more details on which figure the reader should refer to here.

"…with a small and delayed increase of grain index (black line in Fig. 3, in both cases, the main increase of grain index happens after the 15th of January, whereas for normal years, it starts the first week of December). This delayed start of the metamorphism enables the surface snow to retain the enriched summer isotopic composition of precipitation. Nevertheless, a more extensive time series would be necessary to further ascertain this causal relationship."

Page 20, Line 21: "By contrast, there is no apparent relationship between the isotopic composition of precipitation and the grain index from 2008 to 2011." -> Please provide more explanation for this statement because the peaks between the isotopic composition of precipitation and the grain index matches well.

The match between grain index and precipitation is purely casual, as both are linked with temperature. Metamorphism is the coarsening of the snow grain in the firn. Precipitation samples were captured when the snowflakes didn't have any contact with the firn. The sentence has been modified to better express this concept:

"In contrast, the isotopic composition of precipitation was not affected by metamorphism, and there is thus no link between the precipitation isotopic composition and the grain index variations in Fig. 3."

Page 21, Figure 7: The colors in the plot and in the caption are different.

Indeed, we have corrected this issue. The colour was changed in the late part of the writing process to make it more similar to the colour in the Fig. 3.

Page 21, Line 1: "From the 16$^{th}$ of December, we observe … a first decrease of SSA indicating …" -> Due to the large variation in the SSA it is quite hard to say that there is a SSA decrease and how do you explain the high SSA around 20$^{th}$ of December?

The large SSA around the 20$^{th}$ of December is indeed a puzzling event. The results from the previous version of the figure were obtained using data manually taken at Dome C during the 2013/14 summer campaign. Instead, we have changed it by automatic SSA measurements, obtained following the protocol of (Picard et al., 2016). The main difference between the two datasets, is that in the previous figure, we were presenting a small sample of hand taken data gathered over a large area by an observer (not randomly, but in order to cover the largest possible range of conditions) whereas in the new figure (joined in the paper), we present an automated measurement realised by an instrument on a pole of the same area of 38m². This SSA time series is not the one realised in parallel of the isotope sampling, but has the advantage of scanning a larger surface and presenting more homogeneous results as it's always the same surface.

We still observed a peak around the 20$^{th}$ of December, but the values are more within the range of the other observed values. The main interest is that we obtain a time series with smaller uncertainties. The text has been changed in consequence.

Page 21, Line 3: "… numerous drift events mix the snow and therefore cause strong spatial variability." -> Do you have evidence for this conclusion?

Yes, there are on field observations of the drift events. The dates have been included in the manuscript.

"Before 31 December, several drift events mixed the snow and caused a high spatial variability (events observed on 10, 23, 29 December and on 1 January)."

Page 21, Line 4 – 14: -> Please provide more measurements/results to validate this statement. Why is there a sudden drop around 8$^{th}$ of January?

This sudden drop was simultaneous with an observation of large amount of frost deposition. This is not discussed in the manuscript because we only have one occurrence for such an event.

Page 22, Line 1: "… include both spatial and temporal variations as only one sample per day was taken, therefore some of the variability might be due to spatial variability."

-> Does it make sense to use the data if you can have a variation of up to 18 ‰ which is quite large?

The variations of 18permil are not only linked to the spatial variability. The investigation of the amplitude of the spatial variability was realised in section 3.1.2, and estimates that the uncertainty linked to spatial variability with a single sample is 4.8permil (calculated as 2 std). Here, we thus argue that the 18permil signal is real signal (More than 7 standard deviations). This signal was simultaneous with an observation of large amount of frost deposition. This is not discussed in the manuscript because we only have one occurrence for such an event. A note about the spatial variability impact on the measurements has been included at this point:

"A large precipitation event near 2 January is likely the cause of the 18 ‰ increase in the surface snow isotopic composition, which is mirrored in the sub-surface layer about two days later. The associated uncertainty possibly linked to spatial variability was estimated from replicates to be about 4.8 ‰ (2 standard deviations)."

Page 22, Line 28: "… from the annual accumulation at Dome C (7.7 cm)." -> How did you get this value (7.7 cm)?

This is the snow accumulation in snow equivalent and not water equivalent. Before the firnification has occurred, it is more relevant to describe the annual cycles observed in the snow. It has been precised.

Page 22, Line 31: "… spacing between d$^{18}$O$_N$ …" -> What does "N" stand for?

We have clarified why we use this notation:

"… snow isotopic composition $\delta^{18} O_N$ (we call $\delta^{18} O_N$ the isotopic composition

of the snow deeper in the firn, opposed to $\delta^{18} O_S$ which was for the snow surface, see Fig. 1)”

Page 22, Line 31: “…between d$^{18}$ON maxima in the profiles … present a systematic average value of 20 cm” -> I cannot see systematic maxima in these graphs but a variation between 20 cm and 40 cm …

Indeed, the length of the cycles varies between 15 and 40 cm but we observe an average value of 20 cm for each site. This has been precised .The average value across snow pits is systematically 20cm. The emphasis on average has been enhanced (Page 22, line 28):

“While the interpeak distance in individual d18ON profiles varies between 10 and 40 cm, the average spacing between d18ON maxima presented in both Fig. 8 and 9 is consistent across the snowpits and the value is about 20 cm, i.e. considerably different from the expected 8 cm.”

Page 23, Figure 8: At which depth were the snow pits taken? How did you make sure that the snow samples were air-tight, especially from the year 1977 and 1978?

The depths are included in the figure 8, as it is the x-axis. The pits from 77 and 78 were measured in 1979 and remained unpublished until now. Furthermore, for isotopic measurements, air-tightness is not important as the samples kept frozen do not change through time.

Page 23, Line 14: “… Vostok with seven snowpits with …” -> It’s six according to Table 4.

A snowpit has been added in the late part of the writing process and table 4 hasn’t been updated. This has been corrected.

Page 24, Line 1: “… but our manual counting method, applied to a limited number of pits with relatively low resolution, would not enable to detect small differences.” -> What do you mean by “small differences”? What differences?

This has been precised:

“However our manual counting method, in combination with the limited vertical resolution of certain pits makes it difficult to attribute any statistical significance to the small differences shown in Table 4”

Page 24, Line 8: “… of the potential climate signal and non-climate noise.” -> What do you mean by “non-climate noise”? Is it a local signal?

It is a combination of local signal and of different processes which are destroying or modifying the signal. In either case, this signal is different from the original climatic signal recorded in the precipitation. The term is directly taken from (Münch et al., 2017) which is cited here. Precisions have been given:

“As the inter-annual variability in precipitation should be similar across one site, the observed differences must be due to non-climatic (post-deposition) processes, smoothed by diffusion (Munch et al., 2017).”

Page 25, Line 10: “The limited resolution of the S2 profile may thus explain why no seasonal cycle of isotopic composition is visible.” -> Please mention again the resolution of the S2 profile. In this statement, you say that no cycle is visible of the S2 profile but in Figure 12 a cycle of isotopic composition is visible…

This has been precised (Page 25, line 3):
“The limited vertical resolution at S2 of 3 cm may explain why the expected 6 cm seasonal cycle in isotopic composition was not found”

The cycle observed in Figure 12 cannot be linked to the seasonal cycle because it is roughly around 20cm instead of the 6cm expected for the seasonal cycle.

Page 25, Line 28: “… similar to the one found from the data from the transect between Terra Nova Bay and Dome C …” -> Please provide the number.

Included

Page 25, Line 26 – Page 26, Line 2: Please show in Figure 10 all the extracted slope you mention in this section.

It has been tried but it didn't work out. The figure was unreadable. The table 5 presents all the slopes mentioned in this section.

> Page 26, Line 8: "The reduced summer temperature inversion at Dome C is thus not taken into account in the MCIM which could also lead to a reduced slope." -> Does it make sense to compare the Model with measurements? How big is the reduced summer temperature inversion?

The model has been developed to predict the snow isotopic composition variability in Antarctica for this kind of conditions as it attested by multiple papers (Ciais and Jouzel, 1994; Ciais et al., 1995; Landais et al., 2012; Touzeau et al., 2016; Winkler et al., 2012) . We present here some of the limits of the model, which has already been raised before in (Landais et al., 2012). We included a link to this paper for more details as we believe that this is out of the scope of this present manuscript.

> Page 27, Line 15: "As the phase lag is smaller in 2011 …" -> Which "phase lag"?

This has been precised:

> "As the phase lag between surface snow $\delta^{18}O_s$ and temperature was smaller in 2011,"

> Page 27, Line 14 – Page 28, Line 2: Please show the extracted slopes in Figure 10.

Same than the previous comment about the same issue.

> Page 29, Line 1: "(d – excess or $^{17}O$ – excess)" -> I would recommend to change it to "(d$_{excess}$ or $^{17}O_{excess}$)"

The standard notation recommended by the IAEA (see for instance this recent report: https://www-pub.iaea.org/MTCD/publications/PDF/te_1453_web.pdf) is the use of "d-excess" and "$^{17}O$-excess".

We would like to thank the reviewer 2 for all these comments, we hope that we have addressed them all, and we believe that answering them has already improved the quality of the paper.

Casado, M., Landais, A., Masson-Delmotte, V., Genthon, C., Kerstel, E., Kassi, S., Arnaud, L., Picard, G., Prie, F., Cattani, O., Steen-Larsen, H.C., Vignon, E. and Cermak, P. (2016) Continuous measurements of isotopic composition of water vapour on the East Antarctic Plateau. Atmos. Chem. Phys. 16, 8521-8538.

Ciais, P. and Jouzel, J. (1994) Deuterium and oxygen 18 in precipitation: Isotopic model, including mixed cloud processes. Journal of Geophysical Research: Atmospheres 99, 16793-16803.

Ciais, P., White, J., Jouzel, J. and Petit, J. (1995) The origin of present-day Antarctic precipitation from surface snow deuterium excess data. Journal of Geophysical Research: Atmospheres 100, 18917-18927.

Genthon, C., Piard, L., Vignon, E., Madeleine, J.B., Casado, M. and Gallée, H. (2017) Atmospheric moisture supersaturation in the near-surface atmosphere at Dome C, Antarctic Plateau. Atmos. Chem. Phys. 17, 691-704.

Genthon, C., Six, D., Gallée, H., Grigioni, P. and Pellegrini, A. (2013) Two years of atmospheric boundary layer observations on a 45-m tower at Dome C on the Antarctic plateau. Journal of Geophysical Research: Atmospheres 118, 3218-3232.

Genthon, C., Six, D., Scarchilli, C., Ciardini, V. and Frezzotti, M. (2015) Meteorological and snow accumulation gradients across Dome C, East Antarctic plateau. International Journal of Climatology, n/a-n/a.

Landais, A., Ekaykin, A., Barkan, E., Winkler, R. and Luz, B. (2012) Seasonal variations of $^{17}$O-excess and d-excess in snow precipitation at Vostok station, East Antarctica. Journal of Glaciology 58, 725-733.

Merlivat, L. and Jouzel, J. (1979) Global climatic interpretation of the deuterium-oxygen 18 relationship for precipitation. Journal of Geophysical Research: Oceans 84, 5029-5033.

Münch, T., Kipfstuhl, S., Freitag, J., Meyer, H. and Laepple, T. (2017) Constraints on post-depositional isotope modifications in East Antarctic firn from analysing temporal changes of isotope profiles. The Cryosphere 11, 2175.

Picard, G., Libois, Q., Arnaud, L., Verin, G. and Dumont, M. (2016) Development and calibration of an automatic spectral albedometer to estimate near-surface snow SSA time series. The Cryosphere 10, 1297-1316.

Touzeau, A., Landais, A., Stenni, B., Uemura, R., Fukui, K., Fujita, S., Guilbaud, S., Ekaykin, A., Casado, M., Barkan, E., Luz, B., Magand, O., Teste, G., Le Meur, E., Baroni, M., Savarino, J., Bourgeois, I. and Risi, C. (2016) Acquisition of isotopic composition for surface snow in East Antarctica and the links to climatic parameters. The Cryosphere 10, 837-852.

Vignon, E., Wiel, B.J.H.v.d., Hooijdonk, I.G.S.v., Genthon, C., Linden, S.J.A.v.d., Hooft, J.A.v., Baas, P., Maurel, W., Traullé, O. and Casasanta, G. (2017) Stable boundary-layer regimes at Dome C, Antarctica: observation and analysis. Quarterly Journal of the Royal Meteorological Society 143, 1241-1253.

Winkler, R., Landais, A., Sodemann, H., Dümbgen, L., Prié, F., Masson-Delmotte, V., Stenni, B. and Jouzel, J. (2012) Deglaciation records of 17O-excess in East Antarctica: reliable reconstruction of oceanic normalized relative humidity from coastal sites. Clim. Past 8, 1-16.

---

## Referee Report (RR1)

**Archival processes of the water stable isotope signal in East Antarctic ice cores.**

Mathieu Casado et al.

General comments: The revised manuscript shows significant improvements in the structure and it is easier understandable for the reader. The crosslinking between the texts and the figures were significant improved and the individual contents are adequate to get the context. But I would still recommend some small corrections (see specific and detailed comments) before publications.

Specific comment 1: It would be interesting to see in the figures (e.g. figure 3, 5 and 7) the measured wind speed at the location (if available). In the experiments by Ebner et al. 2017, one can see a strong interaction between snow and airflow. Maybe there is also a link between temperature and wind speed on the isotopic composition of the snow.

Detailed comment:

Page 17 Line 12: The font of $n_v \Delta R_v^{18}$ has to be adapted.

Page 21 Line 3: "… index, SSA decreases under the influence of metamorphism)" -> suggest to delete it -> It's the same statement already mention at the beginning of this sentence.

Page 21 Line 4: "On 19 December, it is likely that a precipitation event occurred" -> why "likely"? Based on the Figure 7, there was a precipitation event, also the days before but without a significant influence on the SSA.

Page 23 Line 8: "… to the accumulation rate (around 8 cm)." -> Where is this number from? Maybe I missed it in the manuscript.

---

## Author Response (AR2)

Dear editor,

Please find below the response to the comments from both you and the referees. We would like to thank you and both referees again for the comments on the manuscript which we believe, improved the clarity of the message.

On the behalf of all the co-authors,

Mathieu Casado

**Editor's comments:**

Dear Mathieu

I am pleased to inform you that your manuscript can be published with some minor revisions, as asked by the reviewer. I also suggest for clarity to add a list of author contributions. I am looking forward to your revised version as soon as possible. Best regards Martin Schneebeli Editor TC

We have included author contributions to the manuscript accordingly.

**Referee 2 report:**

General comments: The revised manuscript shows significant improvements in the structure and it is easier understandable for the reader. The crosslinking between the texts and the figures were significant improved and the individual contents are adequate to get the context. But I would still recommend some small corrections (see specific and detailed comments) before publications.

Specific comment 1: It would be interesting to see in the figures (e.g. figure 3, 5 and 7) the measured wind speed at the location (if available). In the experiments by Ebner et al. 2017, one can see a strong interaction between snow and airflow. Maybe there is also a link between temperature and wind speed on the isotopic composition of the snow.

Most of the wind speed data are available. We had previously tried to compare wind speed with the isotopic composition. Comparing the wind speed with the vapour isotopic composition shows interesting results, which were published in Casado et al, 2016 (ACP).

The comparison with the snow isotopic composition is not as straightforward, and we decided not to include them as the results were very difficult to interpret, especially at the seasonal scale (so which would correspond to figure 3). Indeed, the 2-m wind speed presents really strong high frequency variability, and thus, if the hourly data are included in figure 3, it is impossible to see anything. Then, we tried several post-treatment, including comparing the daily average wind speed and the strength of the wind gusts to the isotopic composition. The results were not convincing. For instance, you can see in Fig. R1, the comparison between d18O and wind speed in 2015. The correlation between the two is  $r^2 = 0.03$  (N = 103). In this situation, it is difficult to interpret the results.

This is different for Figure 5 where we only study a 1 day period, and where the exchanges between vapour and snow are directly targeted. As indicated in the paper from 2016 in ACP, we believe that the impact of wind on boundary layer stratification may have a big impact. We have tried to include the wind speed in figure 5 (See Figure R2). The period we are studying is not particularly interesting in this regard as there is no particular signal in the wind speed. This is coherent with the suggestion of a close box as discussed in the manuscript. This has been added in the discussion, but we believe that the wind speed does not bring any additional information and we would rather not include it in figure 5. See Page 14 Line 23:

"The wind speed and direction during this event remained constant, around 2.8 m.s  $^1$  and 165°."

Figure R1: d180 versus wind speed in 2015 at Dome C.

**Figure R2:** Modification of the figure 5 with the wind speed during the study period. The period of interest (between 9pm on 6/01/2015 and 2am on 7/01/2015) does not

exhibit any particular wind speed signal, the value remains at 2.8 m.s-1, which is slightly below the average wind speed (3.3 m.s-1).

We agree with the referee that the wind speed is most likely to be an important variable. It may be important to include it in future studies. Still, we did not manageto make the analogy between the air flow in Ebner et al, 2017 experiment and the surface wind speed in the field. Indeed, the air flow inside the firn is imposed by wind pumping, itself influenced by wind speed but also surface relief which is not well defined in our case. We thus leave this for future studies.

**Detailed comment:**

Page 17 Line 12: The font of nvDRv18 has to be adapted.

**Corrected.**

Page 21 Line 3: "... index, SSA decreases under the influence of metamorphism)" -> suggest to delete it -> It's the same statement already mention at the beginning of this sentence.

We were afraid that the opposite behaviour between grain index presented earlier and SSA presented here could confuse the reader, hence the repeated statement. Nevertheless, it has been deleted as suggested.

Page 21 Line 4: "On 19 December, it is likely that a precipitation event occurred" -> why "likely"? Based on the Figure 7, there was a precipitation event, also the days before but without a significant influence on the SSA.

**It has been deleted.**

Page 23 Line 8: "... to the accumulation rate (around 8 cm)." -> Where is this number from? Maybe I missed it in the manuscript.

We have included the suggested calculation which gives 8 cm in snow equivalent, see page 23, line 6:

"(7.7 cm in snow equivalent, obtained using the values in Table 1 and an average snow density of  $350 \text{ kg m}^{-3}$ , see Table 4)"